# Widespread somatic L1 retrotransposition in normal colorectal epithelium

Chang Hyun Nam[1,10], Jeonghwan Youk[1,2,3,10], Jeong Yeon Kim[2], Joonoh Lim[1,2], Jung Woo Park[4], Soo A Oh[1], Hyun Jung Lee[3], Ji Won Park[5], Hyein Won[1], Yunah Lee[1], Seung-Yong Jeong[5], Dong-Sung Lee[6], Ji Won Oh[7,8], Jinju Han[1], Junehawk Lee[4], Hyun Woo Kwon[9✉], Min Jung Kim[5✉] & Young Seok Ju[1,2✉]

Throughout an individual's lifetime, genomic alterations accumulate in somatic cells[1–11]. However, the mutational landscape induced by retrotransposition of long interspersed nuclear element-1 (L1), a widespread mobile element in the human genome[12–14], is poorly understood in normal cells. Here we explored the whole-genome sequences of 899 single-cell clones established from three different cell types collected from 28 individuals. We identified 1,708 somatic L1 retrotransposition events that were enriched in colorectal epithelium and showed a positive relationship with age. Fingerprinting of source elements showed 34 retrotransposition-competent L1s. Multidimensional analysis demonstrated that (1) somatic L1 retrotranspositions occur from early embryogenesis at a substantial rate, (2) epigenetic on/off of a source element is preferentially determined in the early organogenesis stage, (3) retrotransposition-competent L1s with a lower population allele frequency have higher retrotransposition activity and (4) only a small fraction of L1 transcripts in the cytoplasm are finally retrotransposed in somatic cells. Analysis of matched cancers further suggested that somatic L1 retrotransposition rate is substantially increased during colorectal tumourigenesis. In summary, this study illustrates L1 retrotransposition-induced somatic mosaicism in normal cells and provides insights into the genomic and epigenomic regulation of transposable elements over the human lifetime.

Somatic mutations spontaneously accumulate in normal cells throughout an individual's lifetime, from the first cell division[2–5]. Previous studies on somatic mosaicism have primarily focused on nucleotide variants[6–11]. More complex structural events remain less explored owing, in part, to their relative paucity and technical challenges in detection, particularly at single-cell resolution.

Long interspersed nuclear element-1 (L1) retrotransposons are widespread transposable elements representing approximately 17% of the human genome[12–14]. Evolutionally, L1 retrotransposons are a remarkably successful parasitic unit in the germline through 'copying and pasting' themselves at new genomic sites[15]. However, most of the approximately 500,000 L1s in the human reference genome are unable to transpose further because they are truncated and have lost their functional potential. To date, 264 retrotransposition-competent L1 (rc-L1) sources have been discovered in cancer genomes[16,17] or other experimental studies[12,13,18–21]. Occasionally L1 retrotranspositions have been found in genetic analysis of tissues in several diseases[22,23], implying their role in the development of human diseases and necessitating a more systematic characterization.

Somatic L1 retrotransposition events (soL1Rs) have been systematically explored in cancer tissues[16,17,24]. Specific cancer types, including oesophageal and colorectal adenocarcinomas, showed a higher burden of soL1Rs, which often leads to alteration of cancer genes[17]. In polyclonal normal tissues, soL1R has not yet been clearly studied because it is challenging to detect instances limited to a small fraction of cells. Although several techniques have been previously employed to show soL1Rs in normal neurons, inconsistent soL1R rates have been reported across studies, ranging from 0.04 to 13.7 soL1Rs per neuron[25–30].

To systematically explore soL1R-induced mosaicism in normal cells, we investigated whole-genome sequences of colonies expanded from single cells (hereafter referred to as clones)[2,4]. Our approaches further allowed for simultaneous multi-omics profiling from identical clones[31] and accurate detection of early embryogenic events shared by multiple clones[2,4,5].

## SoL1R in normal colorectal epithelium

In total, we explored 899 whole-genome sequences from clones (Fig. 1a) established from colorectal epithelium (406 clones from 19 donors),

[1]Graduate School of Medical Science and Engineering, Korea Advanced Institute of Science and Technology, Daejeon, Republic of Korea. [2]Genome Insight, Inc., Daejeon, Republic of Korea. [3]Department of Internal Medicine, Seoul National University Hospital, Seoul, Republic of Korea. [4]Korea Institute of Science and Technology Information, Daejeon, Republic of Korea. [5]Department of Surgery, Seoul National University College of Medicine, Seoul, Republic of Korea. [6]Department of Life Science, University of Seoul, Seoul, Republic of Korea. [7]Department of Anatomy, School of Medicine, Kyungpook National University, Daegu, Republic of Korea. [8]Department of Anatomy, Yonsei University College of Medicine, Seoul, Republic of Korea. [9]Department of Nuclear Medicine, Korea University College of Medicine, Seoul, Republic of Korea. [10]These authors contributed equally: Chang Hyun Nam, Jeonghwan Youk. ✉e-mail: hnwoo@korea.ac.kr; minjungkim@snuh.org; ysju@kaist.ac.kr

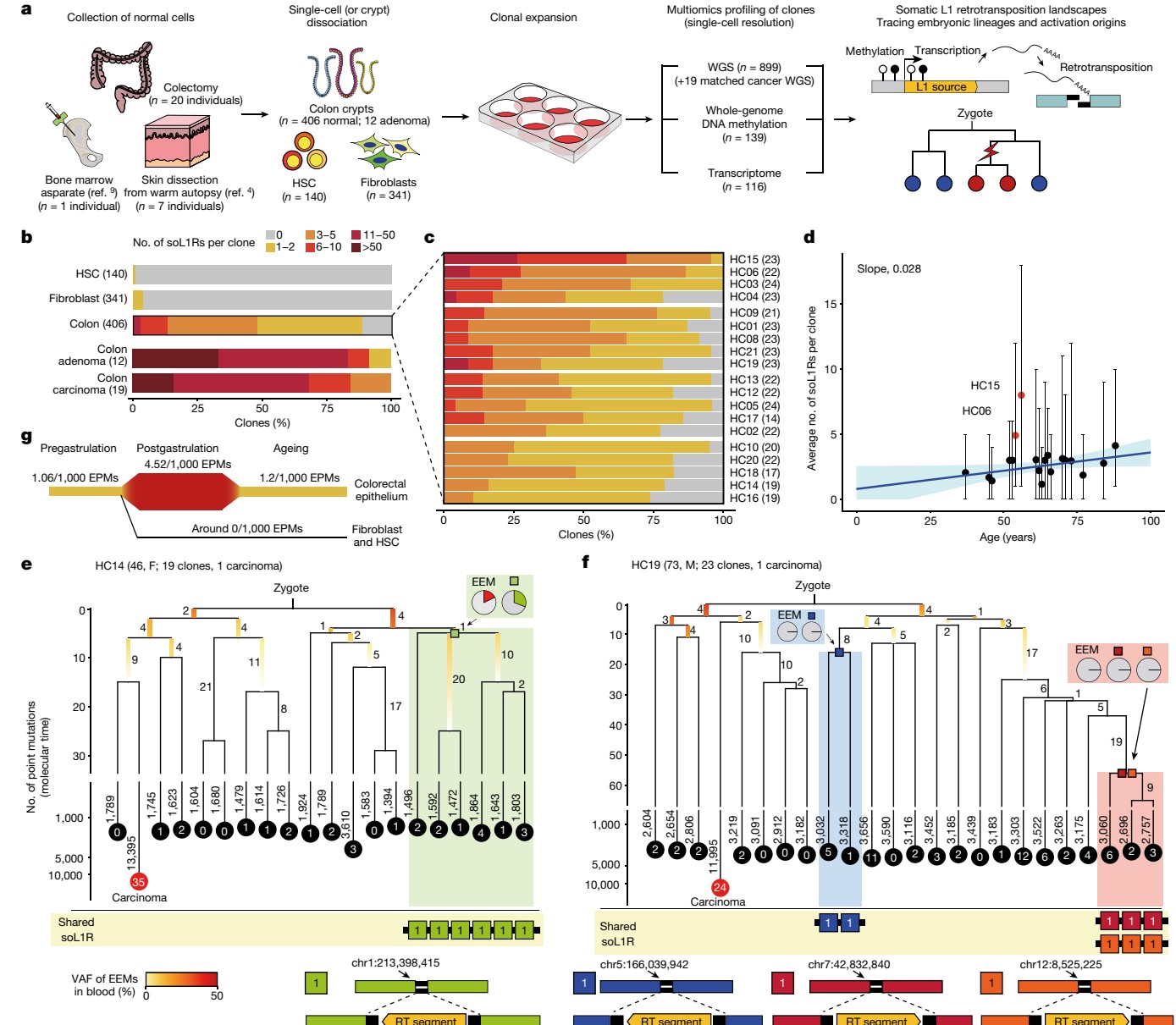

**Fig. 1 | Somatic L1 retrotranspositions in normal cells. a**, Experimental design of the study. HSC, haematopoietic stem and progenitor cells. **b**, Proportion of clones with various numbers of soL1Rs across different cell types (number of clones shown in parentheses). **c**, Proportion of normal colorectal clones with various numbers of soL1Rs across 19 individuals (number of clones shown in parentheses). **d**, Linear regression of the average number of soL1Rs per clone on age in 19 individuals with normal colorectal clones. Vertical line crossing each dot indicates the range of soL1R burden per clone in each individual. Blue line represents the regression line, and shaded areas indicate its 95% confidence interval. Two outlier individuals (HC15 and HC06) are highlighted in red. **e,f**, Early clonal phylogenies of HC14 (**e**) and HC19 (**f**) reconstructed by somatic point mutation. Branch lengths are proportional to the numbers of somatic mutations, which are shown by numbers next to the branches. Early embryonic branches are coloured by variant allele fraction (VAF) of early embryonic mutations (EEMs) in the blood. The numbers of soL1Rs detected are shown in the filled circles at the tips of branches. Pie charts indicate the proportion of blood cells harbouring the EEM or soL1R. RT segment, retrotransposed segment. **g**, Normalized soL1R rates in various stages and cell types.

fibroblasts collected from various locations (341 clones from seven donors)[4], haematopoietic stem and progenitor cells (140 clones from one donor)[9] and MUTYH-associated adenomatous polyps in the colon (12 clones from four polyps of a donor). Additionally we investigated 19 matched colorectal cancer tissues from donors of normal colorectal clones (Supplementary Table 1). From these sequences we assessed somatically acquired mutations, including single-nucleotide variants (SNVs), indels, structural variations and soL1Rs (Supplementary Table 1). These mutations confirmed that the vast majority of the clones were established from a single non-neoplastic founder cell without frequent culture-associated artefacts (Extended Data Figs. 1a,b).

Among the 887 normal and 12 MUTYH-associated adenomatous clones we identified 1,250 and 458 soL1Rs, respectively, by a combined analysis using four different bioinformatics tools (Extended Data Fig. 1c and Supplementary Tables 1 and 2). Of note, soL1R events were clearly distinguished from other genomic rearrangements owing to the two canonical features of retrotransposition—the poly-A tail and target site duplication (TSD; Extended Data Fig. 1d,e). Multiple evidence indicated that most soL1Rs in clones were true somatic events rather than culture-induced events (Supplementary Discussion 1 and Supplementary Fig. 1). In addition, we further found 572 soL1Rs from the 19 matched cancers, 97.2% of which (n = 556) were clonal events shared

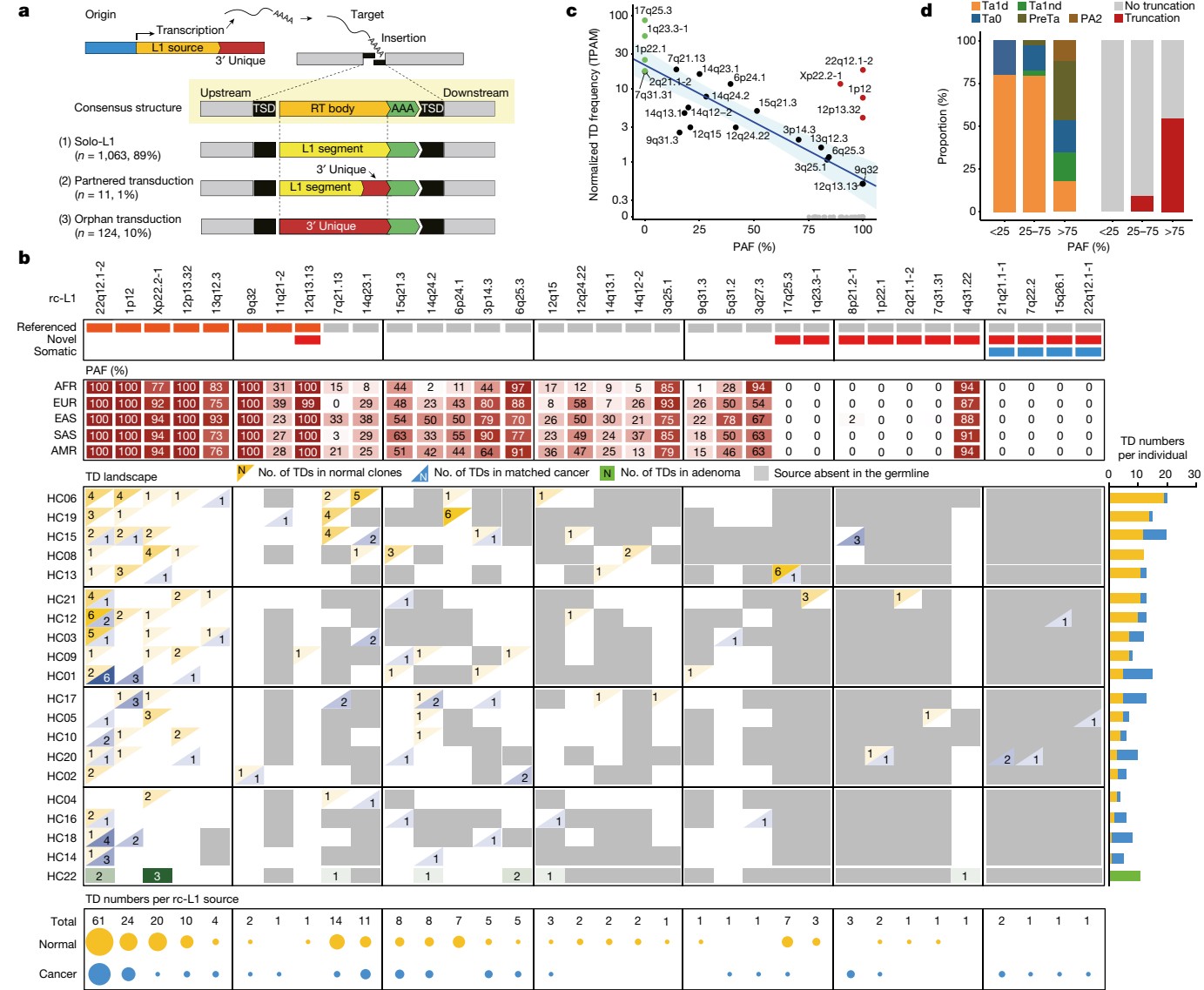

**Fig. 2 | Dynamics of L1 source element activity. a**, Schematic diagram of three classes of L1 retrotransposition: solo-L1, partnered transduction and orphan transduction. **b**, The landscape of transduction events with the features of 34 rc-L1s. TD, transduction; AFR, Africans; EUR, Europeans; EAS, East Asians; SAS, South Asians; AMR, Americans. **c**, Relationship between the population allele frequency of rc-L1s and their normalized retrotransposition activity. Green dots indicate private sources found in just one individual; red dots indicate prevalent-active sources; black and grey dots indicate common sources by all cancer cells in the tissue. For the other retrotransposon types we additionally detected nine somatic Alu insertions in normal clones (Supplementary Table 2).

contributing any and no transduction events in our study, respectively. Blue line represents the regression line of active, but not prevalent, sources and shaded areas indicate its 95% confidence interval. TPAM, number of transductions per L1 allele per 1 million endogenous point mutations of molecular time. **d**, Proportion of L1 subfamily and prevalence of truncating mutations of rc-L1 sources across their PAF. Groups with PAF < 25, 25 < PAF < 75 and PAF > 75 have ten, 34 and 90 L1 sources, respectively.

Of the 1,250 soL1Rs in the 887 healthy clones, 98.9% ($n = 1,236$) were detected from colorectal epithelium, showing extreme cell-type specificity ($P = 9.0 \times 10^{-173}$, two-sided Fisher's exact test). Most normal colorectal clones ($n = 359$, 88%) harboured at least one soL1R, on average three events per clone (Fig. 1b). Remarkably, soL1Rs were more abundant than other classical types of somatic structural variation in clones (Extended Data Fig. 1f).

In colorectal epithelium we found substantial variations in soL1R burden across clones and individuals. The soL1R burden in colorectal clones was between zero and 18 per clone (Fig. 1c). When averaged, soL1R burdens showed a broad but positive relationship with the age of individuals (0.028 soL1Rs per clone per year; Fig. 1d), similar to the

clock-like property of endogenous somatic SNVs and indels (Extended Data Fig. 2a)[32]. This implies that soL1Rs are acquired at a more-or-less constant background rate throughout life in colorectal epithelium. Two outlier individuals further suggest genetic predisposition and/or environmental exposures that stimulate L1 activities (Fig. 1d).

The soL1R burdens were not strongly associated with other features, such as sex and anatomical location of clones in the colon (Extended Data Fig. 2b,c). At the individual clone level, the soL1R burden did not show marked association with other genomic features such as point mutation burden, telomere length, activity of cell-endogenous SNV processes[33] (SBS1 and SBS5/40; standard signatures in the COSMIC database), exposure to reactive oxygen species (SBS18) or colibactin from *pks⁺ Escherichia coli*[34] (SBS88) (Extended Data Fig. 2d–i).

SoL1Rs in normal cells are not confined to the colorectal epithelium, because we detected an additional 37 in 259 laser-capture

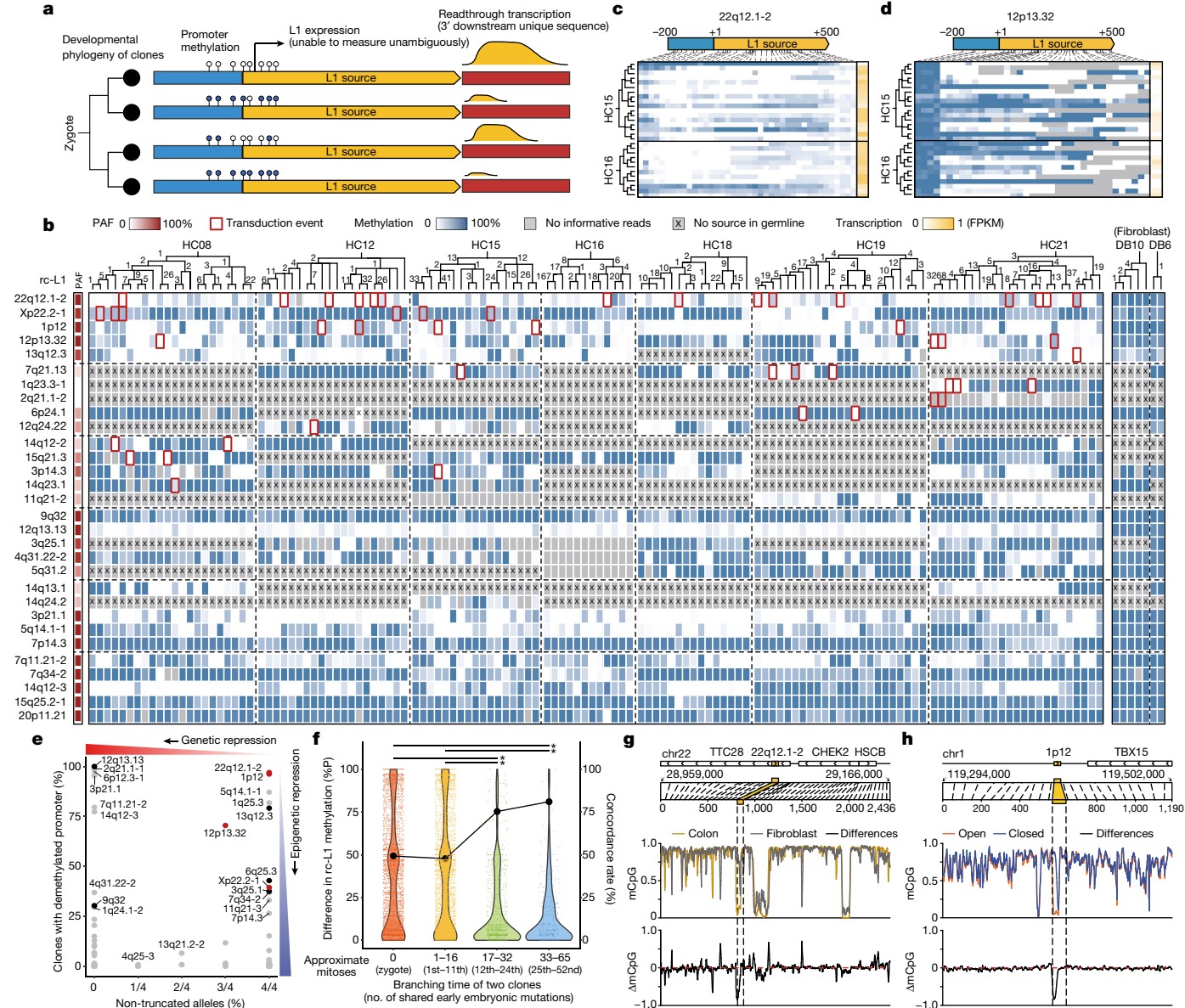

**Fig. 3 | Regulation of L1 source element activity. a**, Schematic diagram of the multidimensional analysis. **b**, Panorama of DNA methylation status of 30 rc-L1s with developmental phylogenies for 132 normal colorectal clones and seven fibroblast clones from nine individuals. It includes 14 rc-L1s contributing any transduction events in these clones and 16 additional rc-L1s showing demethylated promoters in at least five clones. Numbers of branch-specific point mutations are shown in the phylogenies. **c,d**, DNA methylation status and readthrough transcription level of rc-L1 at 22q12.1-2 (**c**) and 12p13.32 (**d**). **e**, Proportion of non-truncation and promoter demethylation of 90 population-prevalent rc-L1s. Red dots, prevalent-active sources; black and grey dots, common sources showing any and no transduction events in our study, respectively.

**f**, Differences in rc-L1 promoter methylation in clone pairs according to their embryonic branching time. The top 30 rc-L1s showing substantial variation in promoter methylation were considered. A fixed mutation rate[4] was used to convert mutation time to embryonic cell generation. %P, percentage point; *$P < 2.2 \times 10^{-16}$ (two-sample Kolmogorov–Smirnov test). **g,h**, Methylation profile of 100 kb upstream and downstream regions of rc-L1 at 22q12.1-2 (**g**) and 1p12 (**h**). The rc-L1 loci are highlighted by yellow rectangles. Top, genomic coordinates and order of CpG sites. Middle, fraction of methylated CpG in colorectal (gold) and fibroblast (silver) clones (**g**), and in colorectal clones with open (orange) and closed (blue) promoters (**h**). Bottom, differences in fraction of methylated CpG depicted in middle panel. mCpG, methylated CpG.

microdissected (LCM) patches from 13 organs[5,11] (Extended Data Fig. 2j and Supplementary Table 3). However, these burdens should not be directly compared to those from colorectal clones because soL1R detection sensitivity is compromised in LCM-based whole-genome sequencing (WGS; Supplementary Discussion 2 and Supplementary Figs. 2 and 3).

## High soL1R activity in embryogenesis

Of the 1,250 soL1Rs in normal clones, 30 were shared by two or more clones in an individual (ten events when collapsed), implying that these events were present in the most recent common ancestral cells of the

clones. Developmental phylogenies of the clones, reconstructed using postzygotic mutations as previously reported[4,5] (Fig. 1e,f and Extended Data Figs. 3 and 4), clearly demonstrated that these soL1Rs were embryonic events. For example, a soL1R event in HC14, shared by six colorectal clones (six out of 19 clones, 32% clonal frequency), was acquired in an ancestral cell at the second-generation node in the phylogeny (Fig. 1e). In addition to the position of the node, the number of postzygotic point mutations ($n = 5$) in the ancestral node supported the idea that the event occurred at the four-cell-stage embryo, given that the first two cell generations in human development generate 2.4–3.8 mutations per cell per cell division (pcpcd) and later cell generations generate

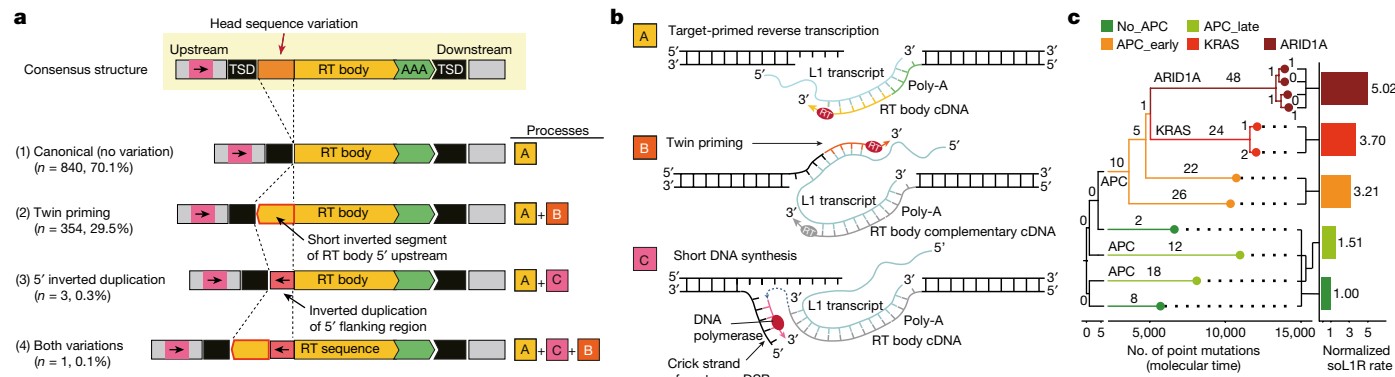

**Fig. 4 | Breakpoint and rate acceleration of somatic L1 retrotranspositions. a,b**, Schematic diagrams of genomic structures of canonical and complex L1 insertions (**a**) and underlying mechanisms (**b**). RT body, retrotransposed body; DSB, double-strand break. **c**, Phylogeny of MUTYH-associated adenomatous clones with normalized L1 rates in groups of lineages classified by driver mutations. Branch lengths are proportional to molecular time, as measured by the number of somatic point mutations. Numbers of branch-specific soL1Rs and branch-specific driver mutations are shown.

0.7–1.2 mutations pcpcd[4,5]. As expected for a pregastrulation event, soL1R was observed in an approximately 200× whole-genome sequence of peripheral blood (mesodermal origin) with around 34% cellular frequency beyond colorectal epithelium (endodermal origin; Fig. 1e). Similarly one somatic Alu insertion, found in HC04, was also probably obtained at the pregastrulation stage (Extended Data Fig. 4).

The other nine shared soL1Rs were probably postgastrulation embryonic events, given the downstream positions and molecular time of their ancestral nodes in the phylogeny (16–56 point mutations of molecular time, equivalent to the 11th–78th cell generations assuming the aforementioned fixed point mutation rate in embryogenesis)[4,5] and the absence of soL1Rs in around 200× whole-genome sequences of blood (Fig. 1f and Extended Data Figs. 3 and 4).

For comparison of various stages and cell types we calculated soL1R rates by counting the number of soL1R events per number of endogenous point mutations[32,33] (EPMs; defined as SBS1 and SBS5/40 SNVs and ID1 and ID2 indels). The soL1R rate in terminal colorectal branches (postdevelopmental colorectal epithelium) was 1.2 per 1,000 EPMs (Fig. 1g). This rate was about four times higher in postgastrulation embryonic branches differentiating to colorectal epithelium (4.52 per 1,000 EPMs, $P = 8.4 \times 10^{-4}$, two-sided Poisson exact test), equivalent to between $1.1 \times 10^{-3}$ and $9.0 \times 10^{-3}$ soL1R pcpcd (assuming a fixed early endogenous point mutation rate[4,5]). Point estimate for the soL1R rate in pregastrulation branches was 1.06 per 1,000 EPMs, although we found one such instance in 28 individuals (Fig. 1e). By contrast, the rates were close to zero per 1,000 EPMs for blood and fibroblast lineages, regardless of embryonic and postdevelopmental stages (Fig. 1g).

## Tracing the source element of soL1Rs

The retrotransposed segments in soL1Rs of normal colorectal clones were mostly the 3' fraction of repetitive L1 sequences ($n = 1,063$, 89%; known as solo-L1; Fig. 2a)[16]. Occasionally the unique downstream sequences of L1 sources were retrotransposed with or without L1 sequences (known as partnered ($n = 11$, 1%) and orphan transductions ($n = 124$, 10%), respectively; Fig. 2a)[16]. In transduction events, fingerprinting of their source elements is possible using the unique sequences as a barcode of L1 sources[16].

Combining colorectal clones and cancer tissues, we found 217 transduction events with 34 L1 sources encompassing these, confirming their retrotransposition competency (Fig. 2b and Supplementary Table 4). Of these, 12 (35%) were new rc-L1s because they did not overlap with 264 active sources previously known[12,13,16–21]. The new rc-L1 sources include three types: (1) one referenced-germline source (present in both the human reference genome and the germline of the individual),

(2) seven non-referenced-germline sources (absent in the reference genome but present in the germline) and (3) four postzygotically acquired sources absent in both the reference genome and germline (Extended Data Fig. 5). Of note, four new non-referenced-germline sources (17q25.3, 1q23.3-1, 1p22.1 and 2q21.1-2; Fig. 2b and Supplementary Table 4) were private to an individual, not being observed in our germline panel encompassing 2,860 individuals from five ancestries. This indicates that the acquisition of new rc-L1 sources is ongoing in the human genome pool, as suggested by population-based genome studies[12,21,35].

## SoL1R activity across source elements

Each of the 34 rc-L1 sources contributed to a different number of transductions in colorectal clones (Fig. 2b). For example, four L1 sources (22q12.1-2, 1p12, Xp22.2-1 and 12p13.32) affected a large fraction (at least 50%) of individuals, causing approximately 50% of the somatic transduction events in our study. These four rc-L1s were prevalent in the population, showing around 100% population allele frequency (PAF) in the human genome pool (Fig. 2b).

Except for these four 'prevalent-active' rc-L1s, high PAF rc-L1s showed low soL1R activity in colorectal epithelium. Most of the 90 rc-L1s with PAF over 75% contributed either none (81, 90%) or one soL1R event (4, 4%) in the 406 colorectal clones. By contrast, rare source elements were often retrotransposed in multiple clones of an individual having the source in the germline. For example, the private source 17q25.3 contributed six events across 22 colorectal clones of HC13 (Fig. 2b).

To compare retrotransposition activities across different source elements, transductions from each rc-L1 were counted per L1 allele per 1 million EPMs of molecular time (referred to as TPAM) in normal colorectal lineages in individuals harbouring the source. Intriguingly, TPAM rates generally showed a negative correlation with the PAF of rc-L1s (Fig. 2c). Rare sources showed higher retrotransposition activities than prevalent sources, except for the four prevalent-active rc-L1s. These features are in line with the inverse relationship between the prevalence and penetrance of human genomic variants[36]. Because rc-L1s can cause insertional mutagenesis, which is potentially damaging, its activity should be repressed through genetic and/or epigenetic mechanisms. Ultrarare sources probably precede sufficient negative selection because they emerged in the human population relatively recently[12].

To understand the genetic foundation of the differential activities across rc-L1s, we explored sequence polymorphisms of the source elements using long-read WGS of two colorectal clones. Population-prevalent source elements were predominantly in the older L1 subfamilies (such as pre-Ta and PA2), as suggested previously[12], and

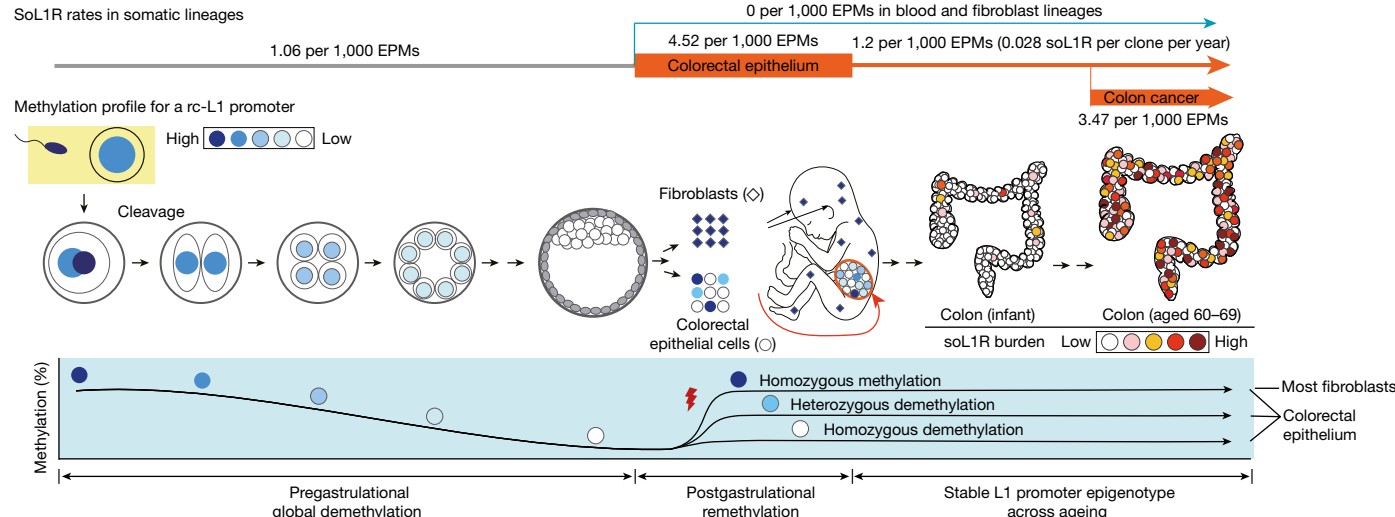

**Fig. 5 | Landscape of somatic L1 retrotranspositions.** Schematic diagram illustrating factors influencing the soL1R landscape. Genetic composition of rc-L1s is inherited from the parents. The methylation landscape of rc-L1 promoters is predominantly determined by global DNA demethylation, followed by remethylation processes in the developmental stages. Then, when an rc-L1 is promoter demethylated in a specific cell lineage, the source expresses L1 transcripts thus making possible the induction of soL1Rs.

harboured open reading frame-disrupting mutations more frequently than rare source elements (Fig. 2d and Supplementary Table 4).

## Dynamics of L1 promoter demethylation

To explore the epigenetic foundation of differential rc-L1 activities in normal cells, we combined whole-genome DNA methylation (in 139 clones) and RNA expression profiles (in 116 clones) in a subset of clones established (Figs. 1a and 3a). As reported for bulk tissues[16,37], these clones represented a strong negative correlation between locus-specific L1 promoter methylation and transcription (Extended Data Fig. 6), suggesting that L1 promoter demethylation is a main switch for L1 transcription.

The frequency of promoter demethylation (and the resultant transcription) varied across cell types and source elements (Fig. 3b, Extended Data Fig. 7, Supplementary Discussion 3 and Supplementary Figs. 4 and 5). Although predominantly methylated in fibroblast clones, rc-L1 promoters were often markedly demethylated in colorectal clones. For instance, promoters of prevalent-active sources 22q12.1-2 and 12p13.32 showed frequent biallelic demethylation (and resultant RNA transcription) in colorectal clones (Fig. 3b–d). Occasionally the clonal frequency of an rc-L1 promoter demethylation was more prevalent in specific individuals, as observed in sources 5q14.1-1 and 14q12-3 (Fig. 3b). Whole-genome DNA methylation profiles from various bulk tissues[38] suggest that colon tissue has a higher frequency of rc-L1 promoter demethylation than any other cell type (Extended Data Fig. 8a).

Of note, we observed that population-prevalent rc-L1s were frequently repressed through promoter methylation and/or genetic truncation. Of the 90 population-prevalent rc-L1s (PAF > 75%), 68 (75.6%) showed predominant promoter methylation in more than 75% of colorectal clones (Fig. 3e). Of the other 22 rc-L1s not preferentially promoter methylated (such as 12q13.13), ten harboured open reading frame-truncating mutations in all informative alleles from the long-read sequencing. The remaining 12 rc-L1s, particularly the four prevalent-active sources (22q12.1-2, Xp22.2-1, 1p12 and 12p13.32), escaped from both genetic and epigenetic repression, which may indicate the functional roles of the sources[39].

Multidimensional analysis further provided four insights into the epigenetic regulation of source elements and subsequent soL1R activity. First, rc-L1 promoter demethylation is a prerequisite condition for soL1Rs. A source element causing any transduction events in a clone was always promoter demethylated in the corresponding clone (Fig. 3b; 47 out of 47, highlighted by red rectangles; 37 homozygous and ten heterozygous demethylations). This further indicates that the demethylated rc-L1 promoter is stable in somatic lineages over time, because its reverse methylation would disrupt such an exclusive association.

Second, the L1 promoter epigenotype is primarily determined in embryogenesis. Autosomal rc-L1 promoter demethylation was predominantly homozygous (Fig. 3b–d), suggesting that it is directly inherited from pregastrulation epigenetic reprogramming, which globally removes DNA methylations in the genome[40–42]. An alternative scenario, stochastic loss of methylation in the ageing process, is less likely because it will preferentially shape demethylation in one allele. Rather, our findings suggest that fully demethylated rc-L1 promoters shaped in the earliest embryonic stage are not sufficiently remethylated subsequently in colorectal epithelial lineages (Fig. 3b). Remethylation should be more thorough in fibroblast lineages because fibroblast clones showed almost complete rc-L1 promoter methylation (Fig. 3b). Molecular time in the clonal phylogenies also indicates that the rc-L1 promoter remethylation process is operational predominantly in the postgastrulation stage. Colorectal clones having their most recent common ancestral cell in the 17–65 embryonic mutations of molecular time (12th–90th cell generations, assuming the above-mentioned fixed early mutation rate[4,5]; near gastrulation to organogenesis) exhibited a higher concordance of promoter epigenotypes for an rc-L1 source (77% concordance rate, 1,446 out of 1,885 clone–L1 pairs) than did clones that diverged earlier (Fig. 3b–d,f).

Third, the range of insufficient remethylation is localized to the promoter of rc-L1 and is independent of other genomic regions. For example, despite the extreme difference in the promoter methylation level of the prevalent-active 22q12.1-2 source between fibroblast and colorectal clones, its 100 kb upstream and downstream regions showed highly similar DNA methylation profiles (Fig. 3g). Likewise, DNA methylation levels of neighbouring and genome-wide regions were largely concordant between colorectal clones, regardless of L1 promoter epigenotype (Fig. 3h and Extended Data Fig. 8b,c).

Last, most L1 transcripts are unproductive regarding soL1Rs in normal cells. A colorectal clone has 17–42 rc-L1 alleles with promoter

demethylation (Fig. 3b), and their transcriptome sequences suggest that a colorectal epithelial lineage is continuously exposed to several rc-L1 transcripts over a lifetime (average 0.6 fragments per kilobase of transcript per million mapped reads (FPKM) when all rc-L1s are aggregated; Extended Data Fig. 7)[43]. However, a clone acquires around three soL1Rs in its lifetime, implying the presence of an active defence mechanism that protects the retrotransposition of L1 transcripts in normal cells.

## Genomic regions of soL1R insertions

The target sites of soL1Rs were broadly distributed genome wide in both normal and cancer cells (Extended Data Fig. 9a). SoL1Rs in normal clones were more frequently inserted in regions of L1 endonuclease target site motifs (190-fold; 95% confidence interval (CI) 78.8–459) and late-replicating regions (5.89-fold; 95% CI 4.48–7.74) as previously observed in cancers[17], although chromatin states and transcriptional levels showed a relatively small effect (Extended Data Fig. 9b).

We observed a substantial level of soL1R depletion in the functional regions of the genome as observed in germline L1s[44]. Among the 1,250 soL1Rs in normal clones we found only one event involving an exon of a protein-coding gene, which showed 29-fold lower frequency than random expectation ($P = 1.9 \times 10^{-11}$, two-sided Poisson exact test). Similarly, soL1Rs were more frequently observed in gene-sparse regions (Extended Data Fig. 9c). SoL1R-combined genomic rearrangements, which represented 1% of soL1Rs in cancer tissues[17], were not observed in normal clones. Our data further demonstrated that soL1R events did not induce additional mutations, gene expression/splicing changes or DNA methylation alterations in nearby regions from retrotransposition sites (Extended Data Fig. 9d–f). We speculate that clones with functionally damaging soL1Rs were negatively selected in normal cells.

## Breakpoints of soL1R events

We further investigated breakpoint sequences at soL1R target sites to infer the mechanistic processes of L1 insertions. In addition to the two canonical features (TSD and poly-A tail), which are acquired by target-primed reverse transcription (process A; Fig. 4a,b), a substantial fraction of soL1Rs showed sequence variations in the 5' head part of the retrotransposed segments, characterized by (1) short inversion in the intraretrotransposed (intraRT) body (n = 354; 29.5%), (2) short foldback inversion (inverted duplication) in the 5′ upstream of the target site (n = 3; 0.3%) or (3) both (n = 1; 0.1%). These sequence variations can be explained by the twin priming mechanism (process B; Fig. 4b)[45] and additional DNA synthesis (around 52–220 base pairs (bp)) potentially by DNA polymerases in the final resolution of L1-mediated insertional mutagenesis (process C; Fig. 4b), respectively. An additional occasional event was observed in a clone established from adenoma, in which part of the precursor mRNA, transcribed in the vicinity of the insertion site, was reverse transcribed and co-inserted into the genome, suggesting strand switching of the reverse transcriptase (Extended Data Fig. 9g). These features collectively illustrate that soL1Rs are not acquired by fully ordered and linear processes, but several optional events can be engaged stochastically[46].

Interestingly, we found two clones, each of which had transductions at different genomic target sites but with exactly the same length of unique sequences (Extended Data Fig. 9h). Given that poly-A tailing is a random event in readthrough transcription, our findings suggest that multiple soL1R events from a single L1 transcript are possible.

## SoL1R acceleration in tumourigenesis

The soL1R burden in the 19 matched colorectal carcinomas showed considerable variance, between four and 105 (Fig. 1b). On average soL1R burden was 30 per cancer, approximately tenfold more frequent than that observed in normal colorectal clones. The soL1R rate in colorectal carcinomas was 3.47 per 1,000 EPMs, which is around threefold higher than in normal colorectal epithelium (Extended Data Fig. 10a). Qualitatively, soL1Rs in tumours shaped more profound changes, including longer insert length (1,031 versus 453 bp for solo-L1, $P = 8.6 \times 10^{-20}$, two-sided $t$-test; 755 versus 615 bp for partnered transductions, $P = 0.59$, two-sided Wilcoxon rank-sum test; and 530 versus 242 bp for orphan transductions, $P = 0.004$, two-sided $t$-test; Extended Data Fig. 10b) and a higher frequency of head sequence variations (41.8 vesus 29.9%, $P = 9.6 \times 10^{-7}$, two-sided Fisher's exact test; Extended Data Fig. 10c). Our findings suggest a permissive condition for L1 retrotransposition in tumour development, not necessarily equivalent to the classical genome instability in cancers. For example, TP53-inactivating mutations and microsatellite and chromosomal instability did not show a robust correlation with soL1R burdens in colorectal cancers (Extended Data Fig. 10d,e). Although chromosomal instability was significant in pancancers encompassing over 2,600 cancer cases[17] (Extended Data Fig. 10f,g), the association was weak and inconsistent in each tumour histologic type (Extended Data Fig. 11).

Acceleration of soL1R rate during tumour development was observed in MUTYH-associated adenomatous clones. In the developmental tree of adenomatous polyps, soL1R rate increased as lineages became closer to carcinoma with an accumulation of more driver mutations. For example, soL1R rate in lineages with three driver mutations (loss-of-function mutations in *APC* and *ARID1A* and a gain-of-function mutation in *KRAS*) was three- to fivefold higher than that in lineages with no marked drivers (Fig. 4c).

## Discussion

Our findings demonstrate that cell-endogenous L1 elements lead to retrotransposition in normal somatic lineages and that colon epithelial cells acquire 0.028 soL1R events per year. Mobilization starts from early human embryogenesis, even before gastrulation, as observed previously[13,47]. The repertoire of rc-L1 is inherited from the parents, and their epigenetic activation is predominantly determined in the postgastrulation embryonic stage, which is then robustly transmitted in the somatic lineage during ageing (Fig. 5). Given the number of crypts in the colon (10 million)[48], individuals in their 60s would collectively have 20 million retrotransposition events in the colorectal epithelium. A small fraction of these L1 insertions can confer phenotypic changes in mutant cells and contribute to human diseases such as cancer[17].

Several complementary methods, including deep sequencing[6], whole-genome amplification[8], duplex DNA sequencing[49], LCM[5,10,11] and in vitro single-cell expansions[2,4,7,9], can be used to explore somatically acquired genomic changes in normal cells. Although clonal expansions are labour intensive and applicable only to dividing cells, they have fundamental advantages[31] including (1) implementation of sensitive and precise mutation detection at the absolute single-cell level, (2) facilitation of additional multi-omics profiling in the same single clones, and (3) permitting the exploration of early developmental relationships of clones.

Although our analyses hint at some mechanisms, many things are yet to be discovered in the dynamics of L1 retrotransposition in normal cells. Owing to their repetitive nature, sequences of source elements and soL1Rs are largely inaccessible by short reads. The mechanistic basis of locus- and cell-type specificity in differential promoter demethylation is puzzling. More comprehensive panoramas on a more significant number of single cells of diverse cell types, from various time points in ageing and disease progression and by more innovative sequencing techniques[50], are warranted to answer these questions.

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

## Methods

### Human tissues

For the in vitro establishment of clonal organoids from colorectal tissues, healthy mucosal tissues were obtained from surgical specimens of 19 patients undergoing elective tumour-removal surgery (Supplementary Table 1). Normal tissues (approximately $1 \times 1 \times 1 \, cm^3$ in size) were cut from a region more than 5 cm away from the primary tumour. Matched blood and colorectal tumour tissues from the same patients were also collected for bulk-tissue WGS.

Fresh biopsies from one patient with MUTYH-associated familial adenomatous polyposis were obtained by colonoscopy. Tissues (approximately $0.5 \times 0.5 \times 0.5 \, cm^3$ in size) were cut from four polyps. Matched blood and buccal mucosa tissue from the same patient were also collected.

All tissues were transported to the laboratory for organoid culture experiments within 8 h of the collection procedure. All procedures in this study were approved by the Institutional Review Board of Seoul National University Hospital (approval no. 1911-106-1080) and KAIST (approval no. KH2022-058), and informed consent was obtained from all study participants. This study was conducted in accordance with the Declaration of Helsinki and its later amendments. No statistical methods were used to predetermine sample size. The experiments were conducted without randomization and the investigators were not blinded during the experimental procedures and data analysis.

### Publicly available datasets

We included publicly available whole-genome sequences of single-cell expanded clones to reach a more complete picture of L1 retrotransposition in various human tissues. We included 474 whole-genome sequences from two previous datasets, one for haematopoietic cells (140 clones from one individual)[9] and one for mesenchymal fibroblasts from our previous work (334 clones from seven individuals)[4]. In addition, we included 259 whole-genome sequences produced from LCM-based patches dissected from 13 organs investigated in a previous study[5,11]. Furthermore, we explored 578 whole-genome sequences generated from LCM-based patches of colorectal tissues[51] to investigate differences in sensitivity for soL1R detection between LCM and clonal expansion methods.

To understand the PAF of rc-L1s we collected 2,852 publicly available whole-genome sequences of normal tissues with known ethnicity information. These data were collected from various studies[52–57].

To understand the impact of the level of genome instability on the frequency of soL1Rs in tumours, we further explored variant calls from the ICGC/TCGA Pan-Cancer Analysis of Whole-Genome (PCAWG) Consortium, which included 2,677 cancer and matched normal whole-genome sequences across around 40 tumour types[17,53]. SoL1Rs from PCAWG samples can be found in a previous paper[17]. Other somatic mutation calls (including TP53-inactivating mutations, structural variations and mutational signatures) generated by the consortium are available for download at https://dcc.icgc.org/releases/PCAWG. Our matrix used in the analysis is available in Supplementary Table 5, which includes driver mutations of 19 matched colorectal cancers identified using CancerVision (Genome Insight).

### Organoid culture of colorectal crypts

All organoid establishment procedures and media compositions were adopted from the literature, with slight modifications[58]. Mucosal tissues were cut into sections of approximately 5 mm and washed with PBS. Tissues were transferred to 10 mM EDTA (Invitrogen) in 50 ml conical tubes, followed by shaking incubation for 30 min at room temperature. After incubation, the tubes were gently shaken to separate crypts from connective tissues. The supernatant was collected, and 20 μl of suspension was observed under a stereomicroscope to check for the presence of crypts. Crypt suspension was centrifuged at 300 relative centrifugal force for 3 min, and the pellet was washed once with PBS to reduce ischaemic time. Isolated crypts were embedded in growth-factor-reduced Matrigel (Corning) and plated on a 12-well plate (TPP). Plating of crypts was performed at limited dilution by modification of the protocol from a previous study[59]. In brief, approximately 2,000 crypts were transferred to 900 μl of Matrigel and $3 \times 150$ μl of droplets were plated in three wells of a 12-well plate. Next, 450 μl of Matrigel was added to the remaining dilution and plating of three droplets in three wells was repeated. Serial dilution was performed at least four times and the final remaining dilution was plated in six wells. Plates were transferred to an incubator at 37 °C for 5–10 min to solidify the Matrigel. Each well was overlaid with 1 ml of organoid culture media, the compositions of which are described in Supplementary Table 6.

### Clonal expansion of single-crypt-derived organoid

Primary culture of bulk and diluted crypts was maintained for at least 10 days to ensure the initial mass of single-crypt-origin organoid. After growth of organoids, a single example was manually picked using a 200 μl pipette under an inverted microscope. The picked organoid was placed in an Eppendorf tube and dissociated using a 1 ml syringe with a 25 G needle under TrypLE Express (Gibco). Next, blocking of TrypLE by ADF+++ (Advanced DMEM/F12 with 10 mM HEPES, 1× GlutaMAX and 1% penicillin-streptomycin) was followed by centrifugation and washing. The pellet was placed in a single well of a 24-well plate. Plates were transferred to a humidified 37 °C/5% $CO_2$ incubator and medium changed every 2–3 days. After successful passage, clonal organoids were transferred to a 12-well plate and further expanded. Confluent clones were collected for nucleic acid extraction and organoid stock.

### Reclonalization of single-crypt-derived organoid

Cultured single-crypt-derived organoids were harvested and dissociated using TrypLE Express. After blocking of TrypLE and washing, organoids were resuspended using ADF+++. Organoid suspensions were filtered through a 40 μm strainer (Falcon), then single cells were sorted into a FACS tube by cell sorter (FACSMelody, BD Biosciences). Single cells were selected based on forward- and side-scatter characteristics according to the manufacturer's protocol. Sorted cells were sparsely seeded with growth-factor-reduced Matrigel (500 per well) in 12-well plates. Grown reclonalized single organoids were manually picked and expanded by the methods described above.

### Primary culture of skin fibroblasts

We obtained seven fibroblast clones for methylation analysis. Dermal skin fibroblasts were cultured by a method described previously[4]. In brief, skin samples were washed with PBS (Gibco) and adipose tissue and blood vessels removed. The remaining tissues were cut into small pieces ($1–2 \, mm^2$) and treated with 1 mg ml$^{-1}$ collagenase/dispase solution (Roche) at 37 °C for 1 h. After treatment, the epidermal layer was separated from the dermal layer and the latter washed with DMEM medium containing 20% FBS (Gibco) to inhibit collagenase/dispase activity. Dermal tissue was then minced into small pieces and cultured in collagen I-coated 24-well plates (Corning) with 200 μl of medium in a humidified incubator at 37 °C with 5% $CO_2$ concentration.

### Library preparation and WGS

For Illumina sequencing we extracted genomic DNA materials from clonally expanded cells, matched peripheral blood and colorectal tumour tissues using either the DNeasy Blood and Tissue kit (Qiagen) or the Allprep DNA/RNA kit (Qiagen) according to the manufacturer's protocol. DNA libraries were generated using Truseq DNA PCR-Free Library Prep Kits (Illumina) and sequenced on either the Illumina HiSeq X Ten platform or the NovaSeq 6000 platform. Colorectal clones were whole-genome sequenced with a mean 17-fold depth of coverage. Matched peripheral blood and colorectal tumour tissues were sequenced with a mean coverage of 181- and 35-fold, respectively. For

PacBio sequencing we extracted genomic DNA from colon organoids using the Circulomics Nanobind Tissue Big DNA kit (Circulomics) according to the manufacturer's protocol. DNA libraries were prepared using the MRTbell express template prep kit 2.0 (PacBio) and sequenced on a PacBio Sequel IIe platform.

### Whole-transcriptome sequencing of organoids
Total RNA was extracted from clonally expanded cells using the Allprep DNA/RNA kit (Qiagen). The total RNA sequencing library was constructed using the Truseq Stranded Total RNA Gold kit (Illumina) according to the manufacturer's protocol.

### Whole-genome DNA methylation sequencing of organoids
Genomic DNA was extracted from clonally expanded cells using either the DNeasy Blood and Tissue kit (Qiagen) or the Allprep DNA/RNA kit (Qiagen). The libraries were prepared from 200 ng of input DNA with control DNA (CpG methylated pUC19 and CpG unmethylated lambda DNA) using the NEBNext Enzymatic Methylation-seq kit (NEB) according to the manufacturer's protocol. Paired-end sequencing was performed using the NovaSeq 6000 platform (Illumina).

### Variant calling and filtering of WGS data
Sequenced reads were mapped to the human reference genome (GRCh37) using the Burrows–Wheeler aligner (BWA)–MEM algorithm[60]. Duplicated reads were removed by either Picard (available at http://broadinstitute.github.io/picard) or SAMBLASTER[61]. We identified SNVs and short indels as previously reported[4]. Briefly, base substitutions and short indels were called using Haplotypecaller2 (ref. 62) and VarScan2 (ref. 63). To establish high-confidence variant sets we removed variants with the following features: (1) 1% or more VAF in the panel of normal, (2) high proportion of indels or clipping (over 70%), (3) three or more mismatched bases in the variant reads and (4) frequent existence of error reads in other clones.

### Calling structural variations
We identified somatic structural variations in a similar way to our previous report[4]. We called structural variations using DELLY[64] with matched blood samples and phylogenetically distant clones to retain both early embryonic and somatic mutations. We then discarded variants with the following features: (1) the presence in the panel of normals, (2) insufficient number of supporting read pairs (fewer than ten read pairs with no supporting SA tag or fewer than three discordant read pairs with one supporting SA tag) and (3) many discordant reads in matched blood samples. To remove any remaining false-positive events and rescue false-negative events located near breakpoints, we visually inspected all the rearrangements passing the filtering process using Integrative Genomics Viewer[65].

### Calling L1 retrotransposition and other mobile element insertions
We called L1 retrotranspositions using MELT[20], TraFiC-mem[16], DELLY[64] and xTea[66] with matched blood samples and phylogenetically distant clones to retain both early embryonic and somatic mutations. Potential germline calls, overlapping with events found in unmatched blood samples, were removed. To confirm the reliability of calls and remove remaining false-positive events we visually inspected all soL1R candidates focusing on two supporting pieces of evidence: (1) poly-A tails and (2) target site duplications using Integrative Genomics Viewer[65]. Additionally we excluded variants with a low number of supporting reads (fewer than 10% of total reads) to exclude potential artefacts. We obtained the 5′ and 3′ ends of the inserted segment to both calculate the size of soL1Rs and determine whether L1-inversion or L1-mediated transduction was combined. When both ends of the insert were mapped on opposite strands, the variant was considered to be inverted. When the inserted segment was mapped to unique and non-repetitive genomic sequences, where a full-length L1 element is located within a 15 kb upstream region, we determined that the L1 insertion was combined with the 3′ transduction and derived from the L1 element on the upstream region of unique sequences. To calculate the VAF of soL1Rs we divided the number of L1-supporting read pairs by the total number of informative read pairs around insertion sites. A read pair was considered informative if the region covering its start and end spanned the insertion breakpoint. Furthermore, we counted the number of reference-supporting read pairs twice when calculating the total number of informative read pairs, because insertion is supported by reads pairs at both ends of the insert. To identify clonal L1 insertions in cancer samples we established a cutoff based on the minimum cell fraction value of shared soL1Rs in normal colorectal clones, because shared soL1Rs are considered true variants. We used the same approach for other mobile element insertions, including Alu and SVA.

### Mutational signature analysis
To extract mutational signatures in our samples we used three different tools (in-house script, SigProfiler[67] and hierarchical dirichlet processes[68]) to achieve a consensus set of mutational signatures for each type of colon sample, including normal epithelial cells, adenoma and carcinoma. In brief, our in-house script is based on non-negative matrix factorization with or without various mathematical constraints, and borrows core methods from the predecessor of SigProfiler[69] such as using a measure of stability and reconstruction error for model selection; however, it provides greater flexibility in examining a broader set of possible solutions, including those that can be missed by SigProfiler, and enables a deliberate approach for determining the number of presumed mutational processes. As a result, we selected a subset of signatures that best explain the given mutational spectrum: SBS1, SBS5, SBS18, SBS40, SBS88, SBS89, ID1, ID2, ID5, ID9, ID18 and IDB for normal colorectal epithelial cells; SBS1, SBS5, SBS18, SBS36, SBS40, ID1, ID2, ID5 and ID9 for MUTYH-associated adenoma; and SBS1, SBS2, SBS5, SBS13, SBS15, SBS17a, SBS17b, SBS18, SBS21, SBS36, SBS40, SBS44, SBS88, ID1, ID2, ID5, ID9, ID12, ID14 and ID18 for colorectal cancers. All signatures are attributed to known mutational signatures available from v.3.2 of the COSMIC mutational signature (available at https://cancer.sanger.ac.uk/cosmic/signatures) and IDB, which is a newly found signature from previous research on normal colorectal epithelial cells[51] but not yet catalogued in COSMIC mutational signature.

### Reconstruction of early phylogenies
We reconstructed the phylogenetic tree of the colonies and the major clone of cancer tissue from an individual by generating an $n \times m$ matrix representing the genotype of $n$ mutations of $m$ samples, as previously conducted[4]. Briefly, SNVs and short indels from all samples of an individual were merged and only variants with five or more mapped reads in all samples were included to avoid incorrect genotyping for low coverage. Additionally, variants with VAF < 0.25 in all samples were removed to exclude potential sequencing artefacts. If the VAF of the $i$th mutation in the $j$th sample was more than 0.1, $M_{ij}$ was assigned 1; otherwise, 0. Mutations shared in all samples were regarded as germline variants and discarded. We grouped all mutations according to the types of samples in which they were found and established the hierarchical relationship between mutation groups. In short, if the samples of mutation group A contain all the samples of mutation group B in addition to other samples, mutation group B is subordinate to mutation group A. We then reconstructed the phylogenetic tree that best explains the hierarchy of the mutation groups. The final phylogenetic tree is a rooted tree in which each sample (colony) is attached to one terminal node of the tree, with the number of mutations in the corresponding mutation group being the length of the branch. For cancer samples, the length of branches represents clonal point mutations with cancer cell fractions greater than 0.7. To convert molecular time (number of early mutations) to physical cell generations we used a mutation rate

of 2.4–3.8 pcpcd for the first two cell divisions and then 0.7–1.2 pcpcd, which were estimated from a previous work[4,5].

### Estimation of soL1R rates in various stages

When calculating soL1R rates we classified point mutations on phylogenetic trees into four different stages: pregastrulation, postgastrulation, ageing (postdevelopment) and tumourigenesis. Mutations shared by multiple clones and detected in bulk blood whole-genome sequences (mesodermal origin) were considered pregastrulational. Mutations in early branches[4,51,70] but not found in bulk blood whole-genome sequences were considered postgastrulational. All other mutations in normal clones were considered to have accumulated during the ageing process. For mutations in ageing and tumourigenesis we counted those attributable to endogenous mutational processes (SBS1 and SBS5/40 for SNVs, ID1 and ID2 for indels), to exclude extra mutations by external carcinogen exposure. For mutations in tumours we counted clonal point mutations (cancer cell fractions greater than 0.7) to exclude subclonal mutations. Finally we calculated soL1R rates in each stage by dividing the number of soL1Rs by the total number of endogenous point mutations. The calculation of soL1R rate for tumourigenesis included only non-hypermutated tumours.

### Population allele frequency of L1 sources

To calculate the PAF of rc-L1 sources we collected 2,852 publicly available and eight in-house (overall 2,860) whole-genome sequences of normal tissues with known ethnicity information (714 Africans, 588 Europeans, 538 South Asians, 646 East Asians and 374 Americans)[52–57]. Initially we determined whether individuals had rc-L1s in their genome. Briefly, we calculated the proportion of L1-supporting reads for non-reference L1 and the proportion of reads with small insert size opposing L1 deletion for reference L1, respectively. Only rc-L1s with a proportion of 15% or more were considered to exist in the genome. We then calculated the PAF of a specific rc-L1 as the proportion of individuals with the L1 in the population.

### Long-read, whole-genome sequence analysis

Sequenced reads were mapped to the human reference genome (GRCh37) using pbmm2 (https://github.com/PacificBiosciences/pbmm2), a wrapper for minimap2 (ref. 71). Sequences for L1-supporting reads near source elements were extracted and mapped to the L1HS consensus sequences[18] using BWA[60]. We next identified sequence variations of source elements, including truncating mutations, and assigned each source element to corresponding L1 subfamilies[21].

### Methylation analysis

Sequenced reads were processed using Cutadapt[72] to remove adaptor sequences. Trimmed reads were mapped using Bismark[73] to the genome combining human reference genome (GRCh37) modified by the incorporation of L1 consensus sequences at the non-reference L1 source sites, pUC19 and lambda DNA sequences. For a single CpG site, the number of reads supporting methylation (C or G), the number of reads supporting demethylation (A or T) and the proportion of former reads among total reads (methylation fraction) were calculated using Bismark. Conversion efficacy was estimated with reads mapped on CpG methylated pUC19 and CpG unmethylated lambda DNA. To observe overall methylation status we examined the methylation fraction in regions ranging from 600 bp upstream to 600 bp downstream from L1 transcription start site for each L1 source element. We then focused on CpG sites located between the L1 transcription start site and the 250 bp downstream region (+1 to +250) and classified each CpG site into one of three categories according to methylation fraction: homozygous demethylation (methylation fraction below 25%), heterozygous (methylation fraction at least 25% and methylation fraction below 75%) and homozygous methylation (methylation fraction at least 75%). Next, methylation scores were assigned to CpG sites (0 for

homozygous demethylation, 5 for heterozygous and 10 for homozygous methylation) and summarized by averaging the score of all CpG sites on the +1 to +250 region of the L1 element. Finally we compared the methylation score across every sample and every known source element to determine the relationship between methylation status and source activation.

For the analysis of L1 promoter methylation level in bulk tissues we downloaded whole-genome bisulfite sequencing data of 16 different tissues from Roadmap Epigenomics[74]. The Roadmap codes are E050 BLD.MOB.CD34.PC.F (Mobilized_CD34_Primary_Cells_Female), E058 SKIN.PEN.FRSK.KER.03 (Penis_Foreskin_Keratinocyte_Primary_Cells_skin03), E066 LIV.ADLT (Adult_Liver), E071 BRN.HIPP.MID (Brain_Hippocampus_Middle), E079 GI.ESO (Esophagus), E094 GI.STMC.GAST (Gastric), E095 HRT.VENT.L (Left_Ventricle), E096 LNG (Lung), E097 OVRY (Ovary), E098 PANC (Pancreas), E100 MUS.PSOAS (Psoas_Muscle), E104 HRT.ATR.R (Right_Atrium), E105 HRT.VNT.R (Right_Ventricle) E106 GI.CLN.SIG (Sigmoid_Colon), E109 GI.S.INT (Small_Intestine) and E112 THYM (Thymus). The methylation fractions of CpG sites in referenced L1 sources were collected and summarized by averaging the fraction of all CpG sites on the +1 to +250 region of the L1 element, then compared the averaged L1 promoter methylation level across different tissues.

### Gene expression analysis

Sequenced reads were processed using Cutadapt[72] to remove adaptor sequences. Trimmed reads were mapped to the human reference genome (GRCh37) using the BWA–MEM algorithm[60]. Duplicated reads were removed by SAMBLASTER[61]. To identify the expression level of each L1 source element we collected reads mapped on regions up to 1 kb downstream from the 3′ end of the source element, and calculated the FPKM value. Only reads in the same direction with the source element were considered. If the source element was located on the gene and both were on the same strand, the FPKM value was not calculated because the origin of reads on the downstream region is ambiguous.

### Association with genome features

The L1 insertion rate was calculated as the total number of soL1Rs per sliding window of 10 Mb, with an increment of 5 Mb. To examine the relationship between L1 insertion rate and other genomic features at single-nucleotide resolution we used a statistical approach described previously[17,75]. In brief, we divided the genome into four bins (0–3) for each of the genomic features, including replication time, DNA hypersensitivity, histone mark (H3K9me3 and H3K36me3), RNA expression and closeness to the L1 canonical endonuclease motif (here defined as either TTTT|R (where R is A or G) or Y|AAAA (where Y is C or T)). By comparison of breakpoint sequences with the L1 endonuclease motif, we assigned genomics regions with more than four (most dissimilar), three, two and fewer than one (most similar) mismatches to the L1 endonuclease motif into bins 0, 1, 2 and 3, respectively. DNA hypersensitivity and histone mark data from the Roadmap Epigenomics Consortium were summarized by averaging fold-enrichment signal across eight cell types. Genomic regions with fold-enrichment signal lower than 1 belonged to bin 0, and the remainder were divided into three equal-sized bins: bin 1 (least enriched), bin 2 (moderately enriched) and bin 3 (most enriched). RNA sequencing data were also obtained from Roadmap and FPKM and averaged across eight cell types. Regions with no expression (FPKM = 0) belong to bin 0 and the remainder were divided into three equal-sized bins: bin 1 (least expressed), bin 2 (moderately expressed) and bin 3 (most expressed). Replication time was processed by averaging eight ENCODE cell types, and genomic regions were stratified into four equal-sized regions: bin 0 contained regions with the latest replicating time and bin 3 contained regions with the earliest replicating time. For every feature, enrichment scores were calculated by comparison of bins 1–3 against bin 0. Therefore, the log value of the enrichment score for bin 0 should be equal to 0 and is not described on plots.

## Reporting summary

Further information on research design is available in the Nature Portfolio Reporting Summary linked to this article.

## Data availability

Whole-genome, DNA methylation and transcriptome sequencing data are deposited in the European Genome-phenome Archive with accession no. EGAS00001006213 and are available for general research use. The human reference genome GRCh37 is available at https://www.ncbi.nlm.nih.gov/data-hub/genome/GCF_000001405.13.

## Code availability

In-house scripts for analyses are available on GitHub (https://github.com/ju-lab/colon_LINE1).

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

**Acknowledgements** We thank S. Park, R. Kim, B.-K. Koo and G. J. Faulkner for their comments and discussions. This work was supported by the National Research Foundation of Korea funded by the Korean Government (nos. NRF-2020R1A3B2078973 to Y.S.J. and NRF-2021R1G1A1009606 to H.W.K.); a grant from the MD-PhD/Medical Scientist Training Programme through the Korea Health Industry Development Institute, funded by the Ministry of Health & Welfare of Republic of Korea; and by the Suh Kyungbae Foundation (no. SUHF-18010082 to Y.S.J.).

**Author contributions** J.Y. and Y.S.J. conceived the study. J.Y., H.W.K., J.Y.K., H.W. and Y.L. developed the entire protocol of clonal expansion of colorectal epithelial cells and conducted experiments. H.J.L., Ji.W.P., S.-Y.J. and M.J.K. collected colorectal samples and clinical histories from patients. S.A.O. conducted genome sequencing. C.H.N. and J.Y. conducted most genome and statistical analyses, with contributions from J.Lim, H.W.K. and Y.S.J. Ju.W.P. and J.Lee contributed to large-scale genome data management. D.-S.L., J.W.O. and J.H. participated in data interpretation. C.H.N., H.W.K. and Y.S.J. wrote the manuscript with contributions from all authors. Y.S.J. supervised the overall study.

**Competing interests** Y.S.J. is a cofounder and chief executive officer of Genome Insight, Inc. The remaining authors declare no competing interests.

**Additional information**
**Correspondence and requests for materials** should be addressed to Hyun Woo Kwon, Min Jung Kim or Young Seok Ju.

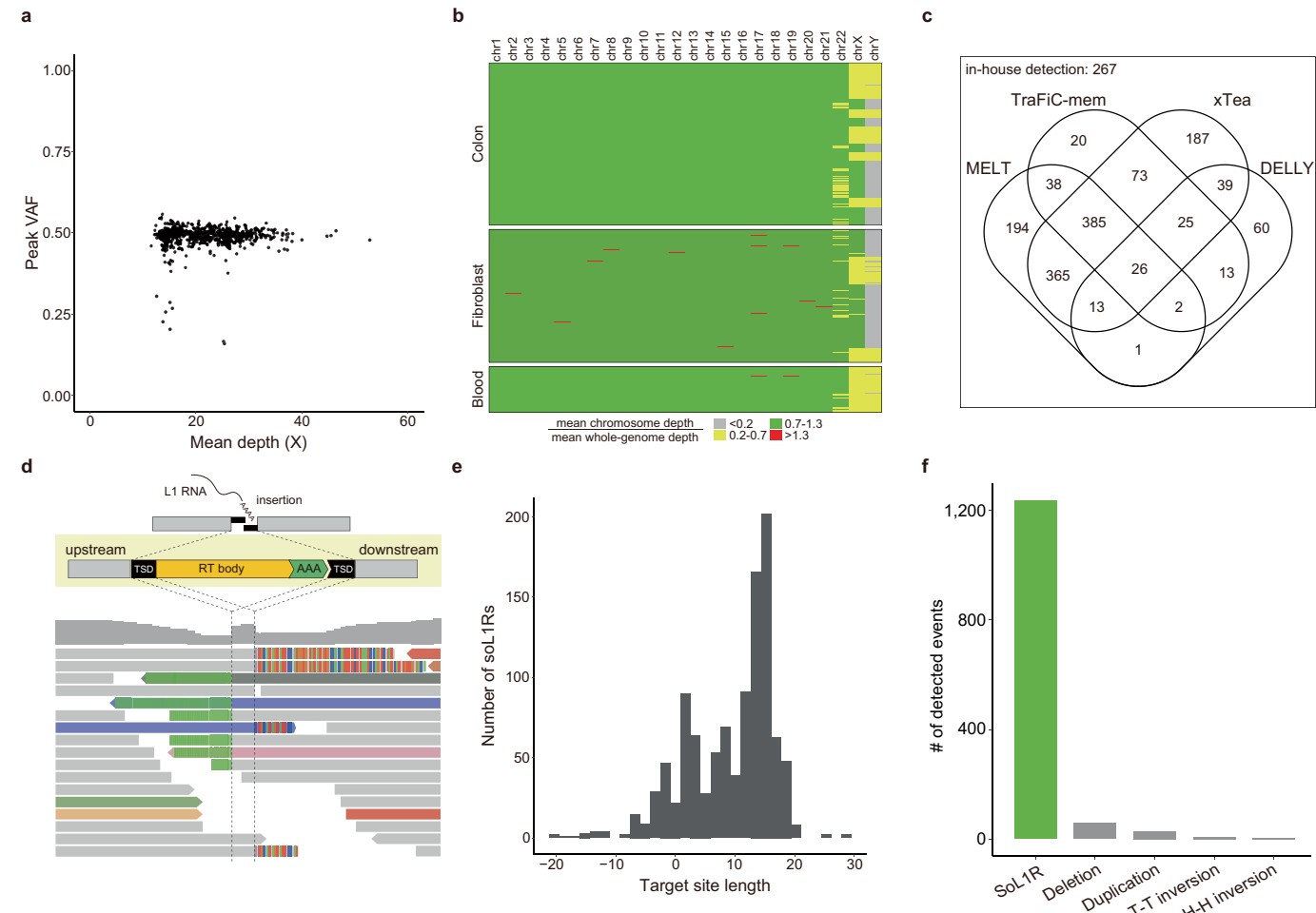

**Extended Data Fig. 1 | Clones for detection of soL1Rs. a**, A scatter plot showing mean sequencing coverage of clones and peak VAF of somatic mutations. Most clones showed their peak VAFs around 0.5, indicating that they were established from a single founder cell. **b**, Chromosome level copy number changes of the 887 normal clones. No significant genome-wide aneuploidy was detected, supporting genomic stability during clonal expansion of normal single cells. **c**, A Venn diagram showing the number of soL1Rs detected by each bioinformatics tool. **d**, A schematic plot describing two genomic footprints of retrotransposition, the poly-A tail and target site duplication. RT body, retrotransposed body; TSD, target site duplication. **e**, The distribution of target site lengths at insertion sites. Positive and negative target site lengths indicate target site duplication and deletion, respectively. soL1R, somatic L1 retrotransposition. **f**, Number of structural variations in 406 clones from normal colon epithelial cells. T-T inversion, tail-to-tail inversion; H-H inversion, head-to-head inversion.

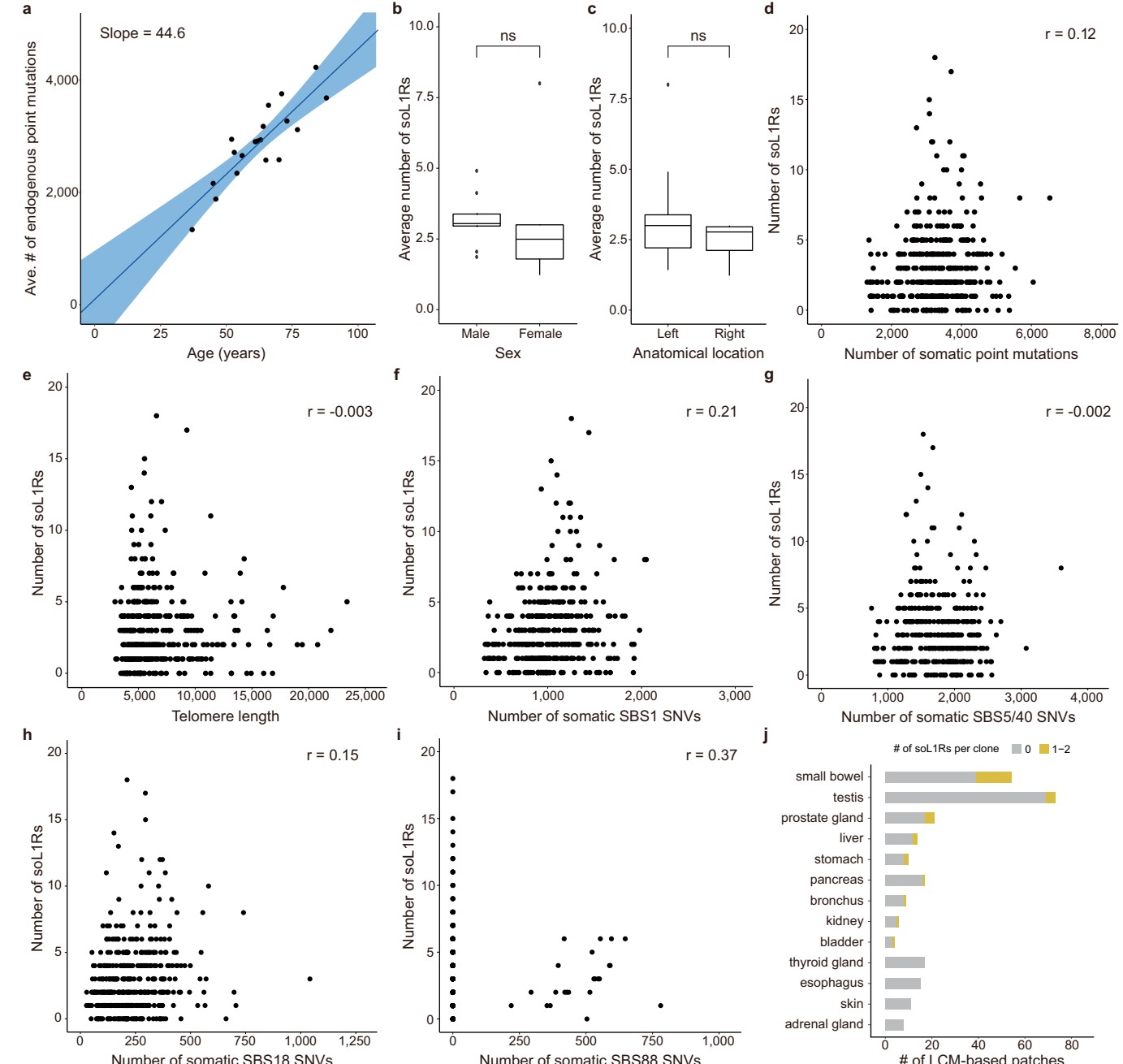

**Extended Data Fig. 2 | Associations between soL1R burden and other genomic features of clones. a**, Linear regression between the average number of endogenous point mutations in the colorectal clones and the age of sampling in 19 individuals. Blue line represents the regression line (44.6 point mutations per year), and shaded areas indicate its 95% confidence interval. The rate is consistent with the rate previously estimated in the colon (43.6 mutations per year from Lee-Six et al., Ref. 51). **b,c**, Comparison of the average number of soL1Rs per individual across sex (b) and anatomical location of the colorectal crypts (c) in 19 individuals with normal colorectal clones with two-sided Wilcoxon rank-sum test. Box plots illustrate median values with interquartile ranges (IQR) with whiskers (1.5 x IQRs). ns, not significant. **d–i**, Relationship between the number of soL1R for each colorectal clone and the number of somatic point mutations (d), telomere length (e), the number of somatic SBS1 SNVs (f, clock-like mutations by deamination of 5-methylcytosine), the number of somatic SBS5+SBS40 SNVs (g, clock-like mutations by unknown process), the number of somatic SBS18 SNVs (h, possibly damage by reactive oxygen species), and the number of somatic SBS88 SNVs (i, damage by colibactin from *pks+ E. coli*). No obvious association was found. **j**, Number of LCM-based patches with various numbers of soL1Rs across different organs. LCM, laser-capture microdissection.

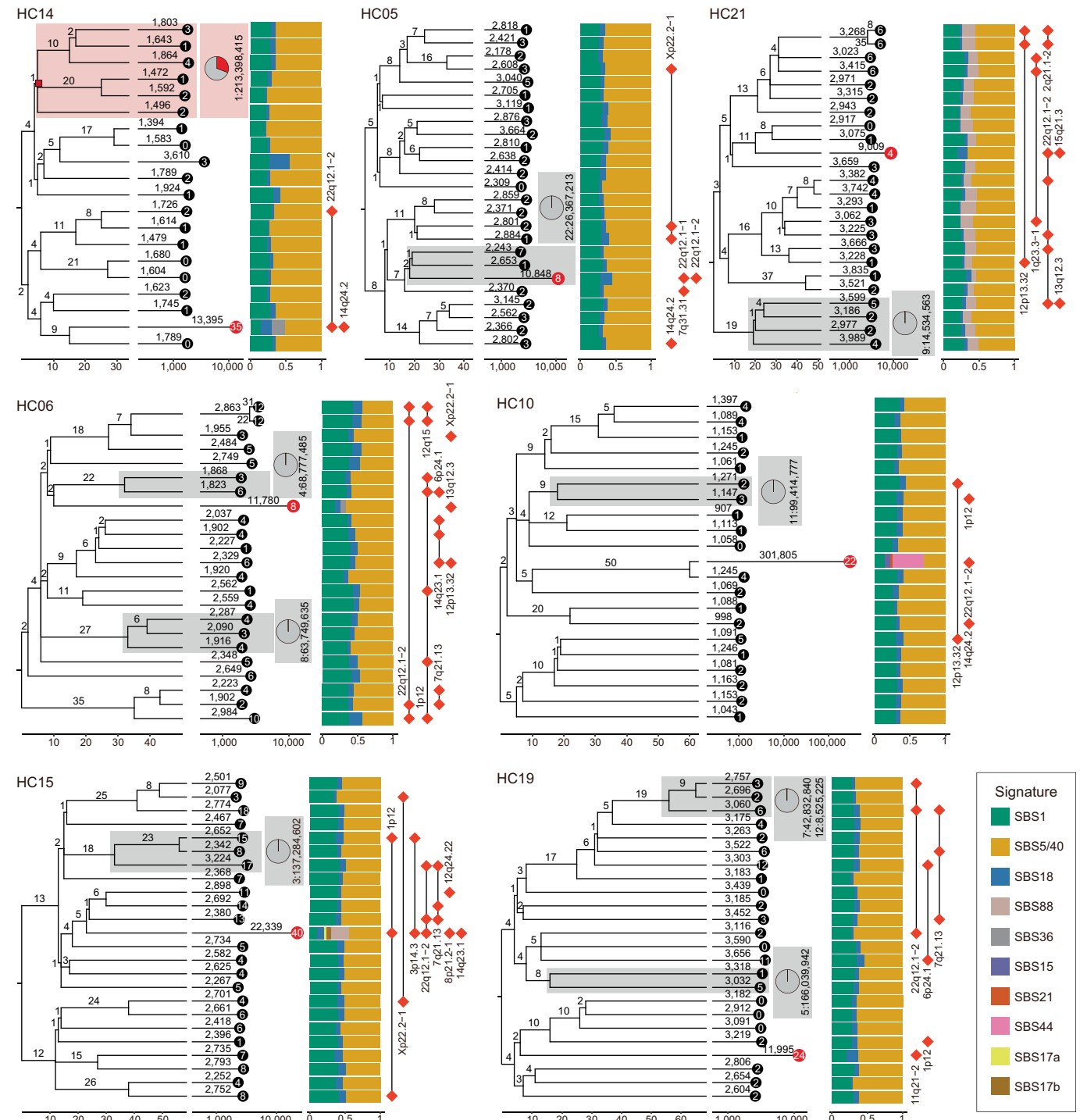

**Extended Data Fig. 3 | SoL1Rs on the developmental phylogenies of the clones from the seven individuals with early embryonic soL1R events.** Early phylogenies of colorectal clones and the matched cancer tissue are shown in seven individuals who have shared soL1Rs among clones. Branch lengths are proportional to the molecular time measured by the number of somatic point mutations. The numbers of branch-specific point mutations are shown with numbers. The filled circles at the ends of branches represent normal clones (black-filled circles) and cancer clones (red-filled circles). The numbers within the filled circles show the number of soL1Rs detected from the clones. Shaded areas indicate somatic lineages with shared soL1Rs. The genomic location of the shared soL1R insertions and the proportion of the blood cells carrying the soL1Rs are shown by genomic coordinates and pie charts. Coloured bars on the right side represent the proportion of mutational signatures attributable to the somatic point mutations. Orange diamonds show L1 sources (origin), which caused transduction events across the colorectal clones.

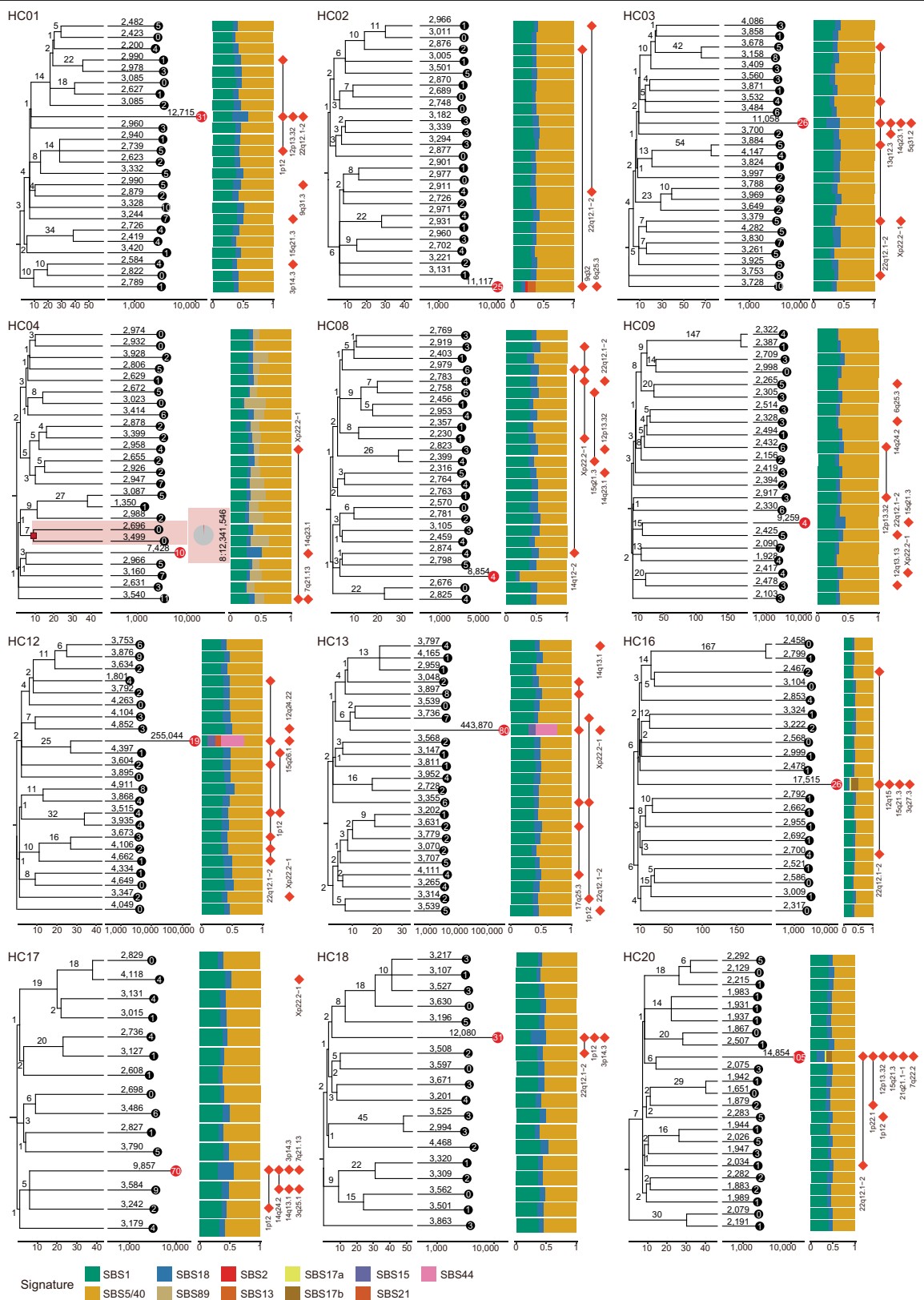

**Extended Data Fig. 4 | SoL1Rs on the developmental phylogenies of the clones from the 12 individuals without early embryonic soL1R events.** Early phylogenies of colorectal clones and the matched cancer tissue are shown in 12 individuals who have no shared soL1Rs among clones. Branch lengths are proportional to the molecular time measured by the number of somatic point mutations. The numbers of branch-specific point mutations are shown with numbers. The filled circles at the ends of branches represent normal clones (black-filled circles) and cancer clones (red-filled circles). The numbers within

the filled circles show the number of soL1Rs detected from the clones. Shaded area indicates somatic lineages with shared Alu insertion. The genomic location of the shared Alu insertion and the proportion of the blood cells carrying the Alu insertion are shown by genomic coordinates and a pie chart. Coloured bars on the right side represent the proportion of mutational signatures attributable to the somatic point mutations. Orange diamonds show L1 sources (origin), which caused transduction events across the colorectal clones.

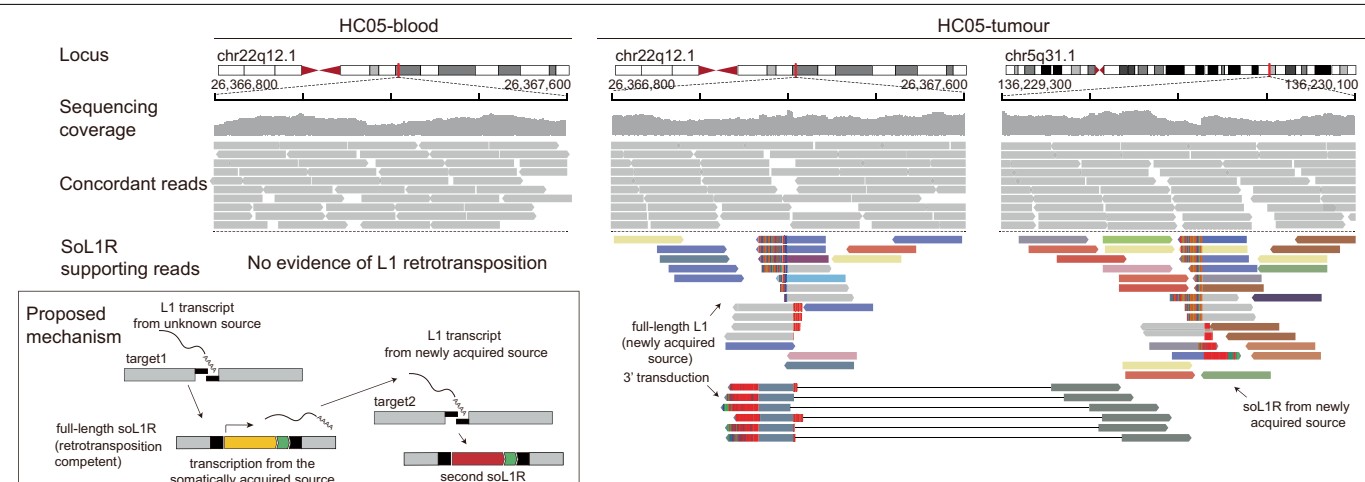

**Extended Data Fig. 5 | An example of a soL1R event induced from a somatically acquired L1 source.** HC05 tumour has a rc-L1 in 22q12.1 (middle) which is not found in the germline of HC05 (blood; left). The rc-L1 (22q12.1-1) caused a transduction event at 5q31.1 (right) in the tumour, suggesting secondary transduction from the new somatically acquired rc-L1. The proposed order of events is summarised in the lower-left panel. SoL1R, somatic L1 retrotransposition.

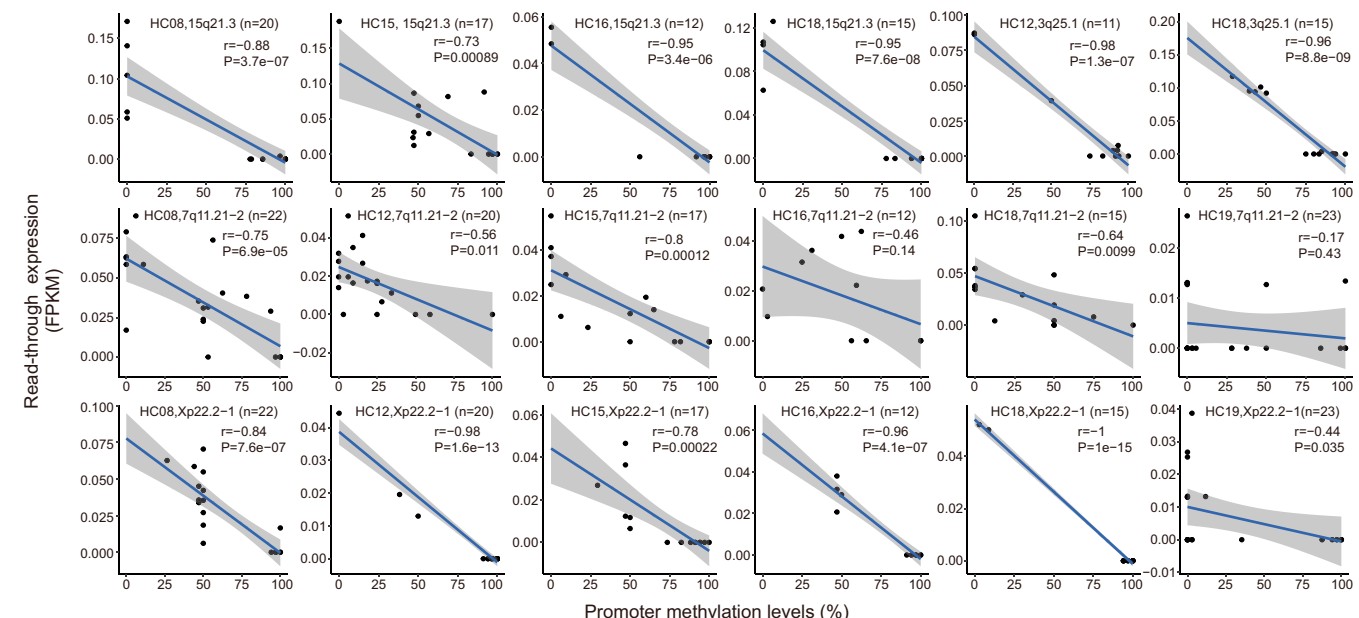

**Extended Data Fig. 6 | Relationship between DNA methylation status and readthrough RNA expression levels.** Relationship between DNA methylation status in the promoter region and readthrough RNA expression level of rc-L1s, which have variable methylation and expression levels, is described in each individual. It only includes cases where there are more than 10 clones with information on methylation and expression levels for a specific rc-L1 in an individual. Correlation coefficient and P-value from Pearson's test is described. Blue line represents the regression line, and the shaded areas indicate its 95% confidence interval. FPKM, fragments per kilobase of transcript per million.

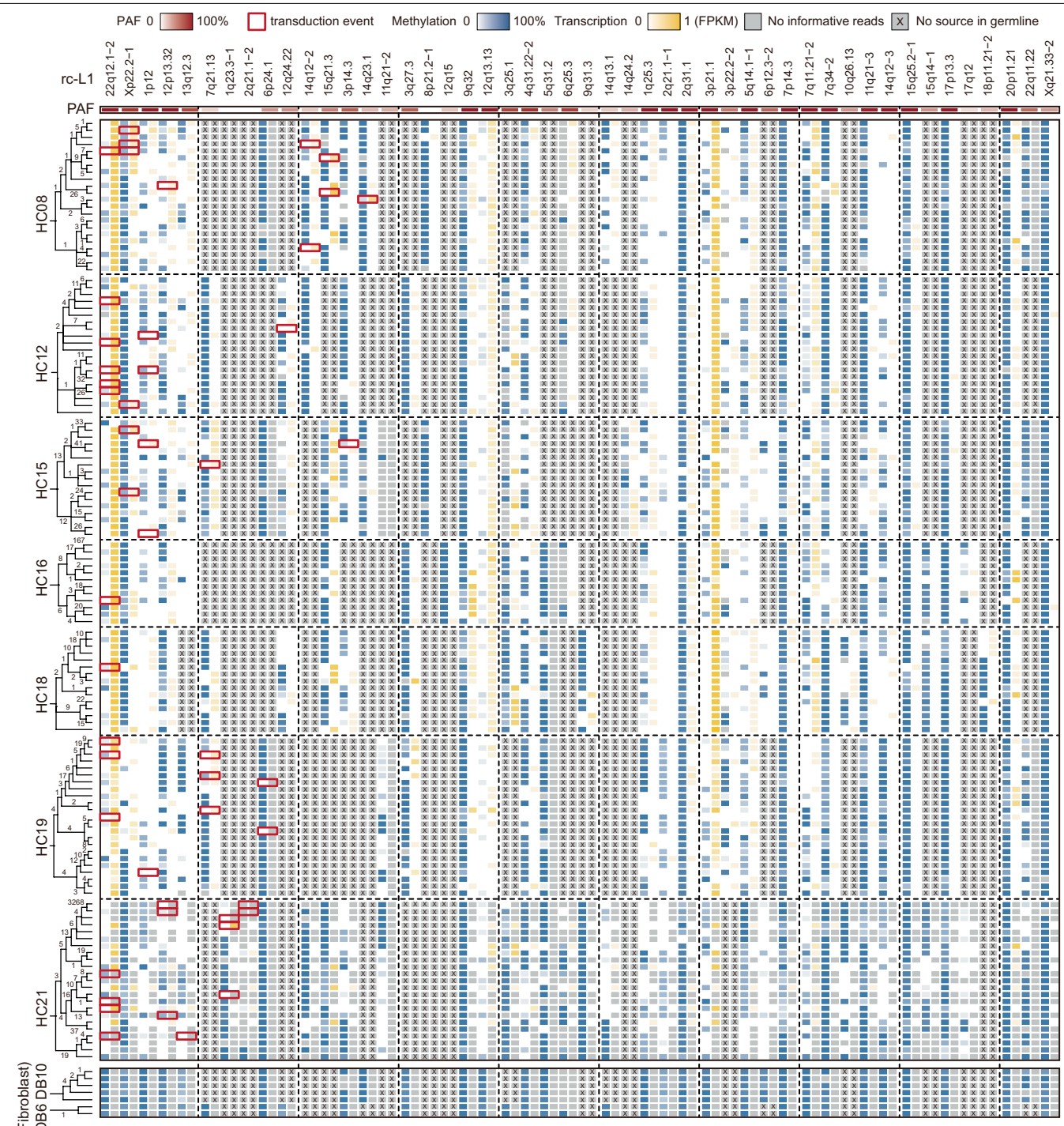

**Extended Data Fig. 7 | Panorama of DNA methylation status and readthrough RNA expression levels of 48 rc-L1s.** DNA methylation status, readthrough RNA expression levels, and developmental phylogenies of 48 rc-L1s in 132 normal colorectal clones and 7 fibroblast clones from 9 patients are displayed. It includes 27 rc-L1s that were active in our colorectal cohort and 21 rc-L1s that harbour demethylated promoters in at least five colorectal clones. The phylogenies are shown on the left side with the number of point mutations (molecular time). PAF, population allele frequency; FPKM, fragments per kilobase of transcript per million; rc-L1, retrotransposition-competent L1.

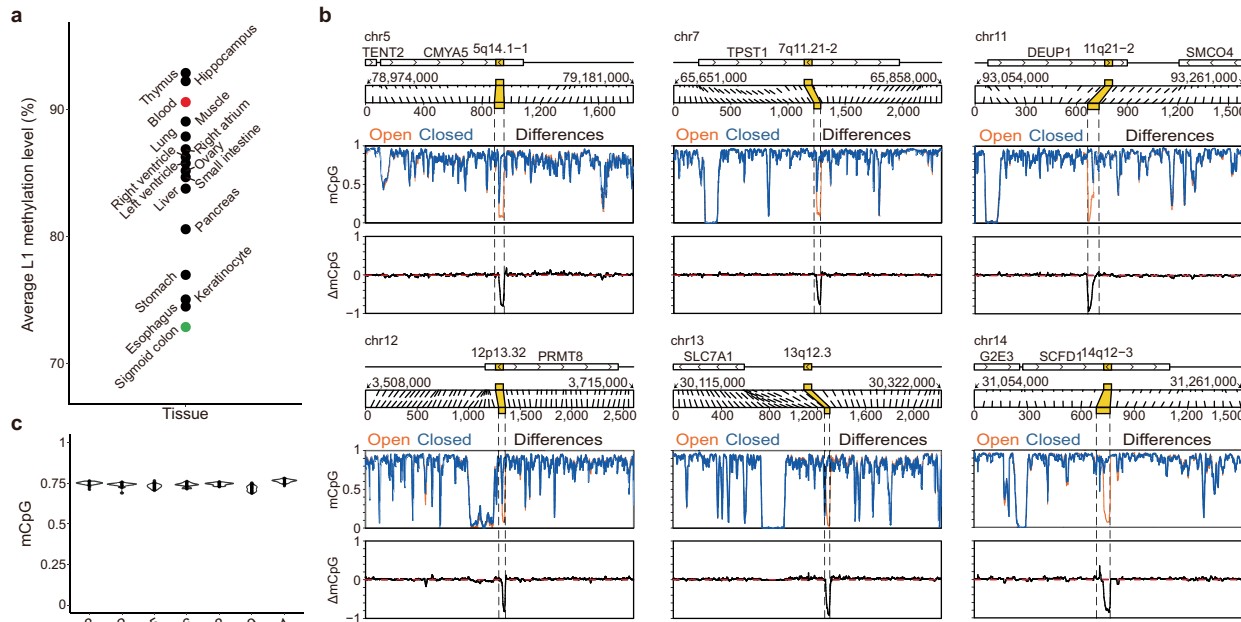

**Extended Data Fig. 8 | DNA methylation levels in various tissues and differences in DNA methylation in the regions nearby source elements and whole-genomes of colorectal clones. a**, Average level of L1 promoter DNA methylation across different tissues from ENCODE. Among 30 rc-L1s described in Fig. 3b, only 12 rc-L1s with sufficient reads in all tissues were selected. **b**, Methylation profiles of 100 kb upstream and downstream regions of 6 reference rc-L1s with variable methylation levels in colorectal clones. The region for rc-L1 is highlighted with yellow boxes. The genomic coordinates and order of CpG sites are depicted in the top panel. Middle panel shows the fraction of methylated CpG in colorectal clones with open (orange) and closed (blue) promoters. Bottom panel shows the differences in the fraction of methylated CpG depicted in the middle panel. mCpG, methylated CpG. **c**, A scatter plot showing genome-wide methylation levels of normal colorectal clones in each individual.

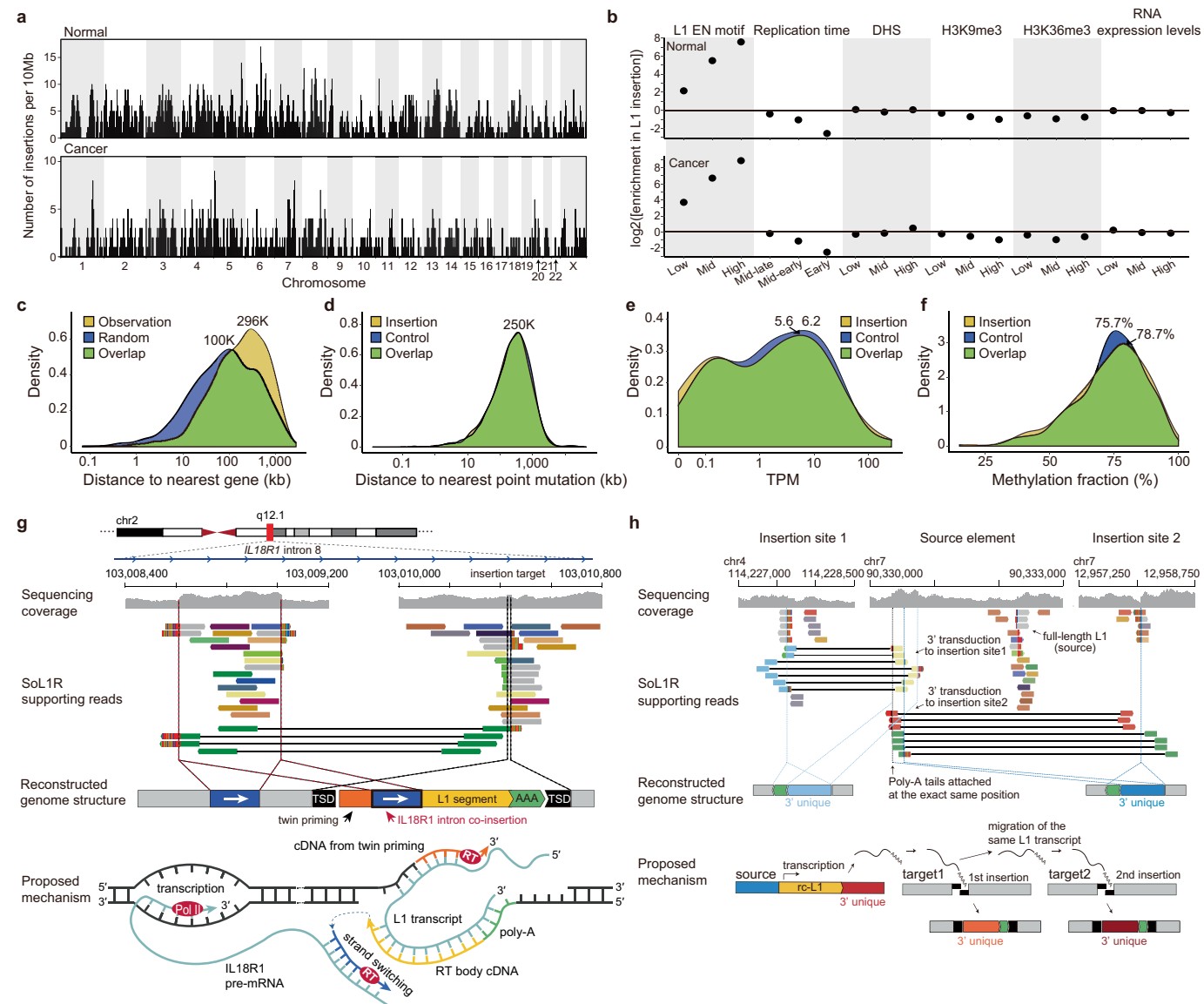

**Extended Data Fig. 9 | Characteristics of soL1R insertion sites and examples of soL1Rs providing insights into the L1 dynamics in somatic cells. a**, Genome-wide distribution of soL1R target sites in normal colorectal clones and 19 matched colorectal cancers. Bars represent the number of L1 insertions in a 10 Mb sliding window with a 5-Mb-sized step. **b**, Association between L1 insertion rate and various genomic features. Dots represent the log value of enrichment scores calculated by comparing bins 1–3 against bin 0 for each feature. L1 EN motif, L1 endonuclease target motif; DHS, DNase I hypersensitivity site. **c**, Distribution of distances to the nearest gene from L1 insertion sites and those from random sites. **d**–**f**, Distribution of distances to the nearest point mutations (d), gene expression level (e), and methylation fraction of nearby region (f) in colorectal clones with and without L1 insertions. TPM, transcripts per million. **g**, An example of a soL1R co-inserted with an expressed gene in the vicinity of the insertion site. A suggestive mechanism is shown in the bottom panel. SoL1R, somatic L1 retrotransposition; TSD, target site duplication; RT body, retrotransposed body. **h**, An example of a clone with two transduction events at different genomic target sites but with the same length of the unique sequences. A suggestive mechanism is shown in the bottom panel.

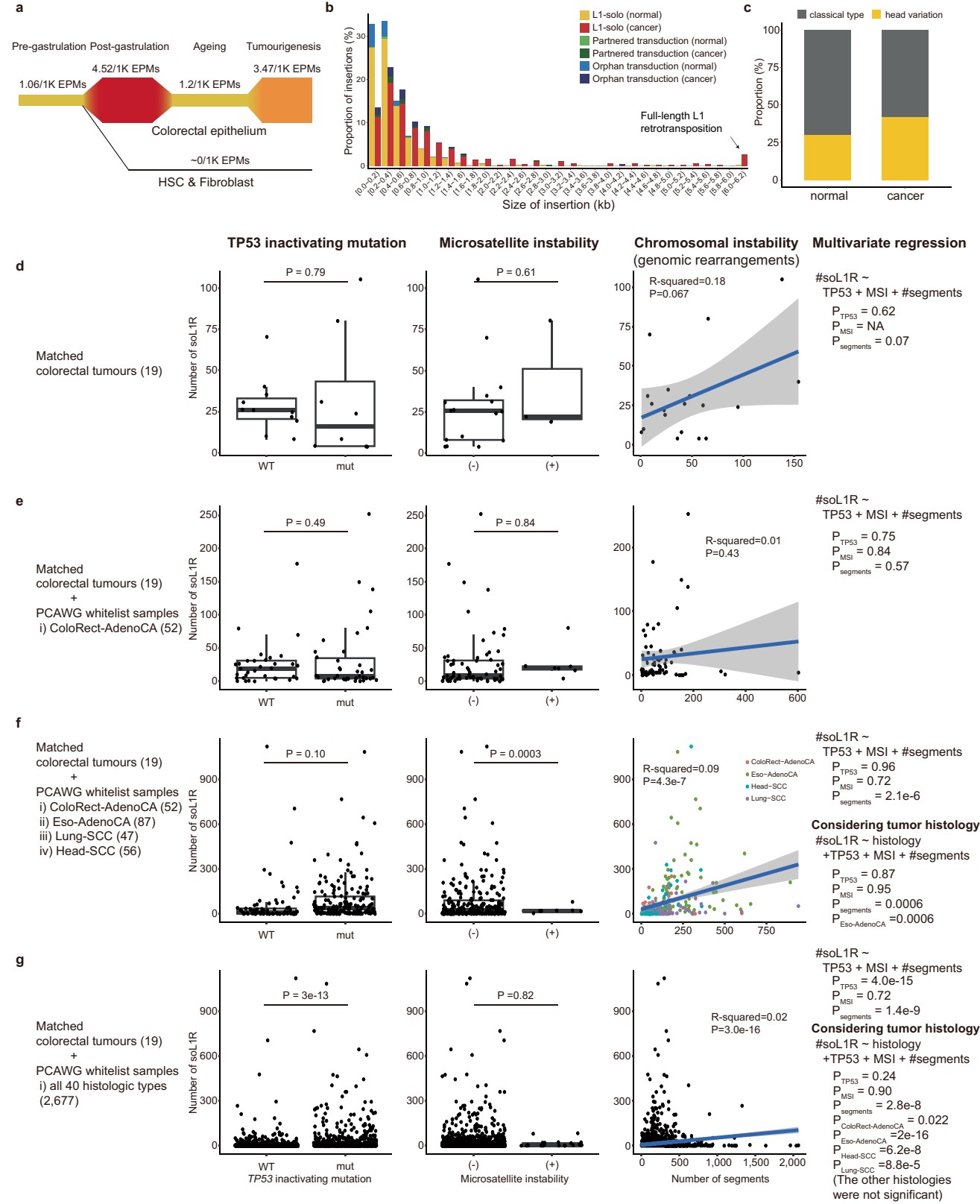

**Extended Data Fig. 10** | See next page for caption.

**Extended Data Fig. 10 | Differences in genomic features of somatic L1 insertion between normal colorectal clones and colorectal cancers.** **a**, The soL1R rate is accelerated during tumourigenesis in colorectal lineages. EPM, endogenous point mutation. **b**, Distribution of L1 insertion size in 406 normal colorectal clones and 19 matched colorectal cancers. **c**, Proportion of soL1Rs events with head variations in 406 normal colorectal clones and 19 matched colorectal cancers. **d**–**g**, The number of soL1Rs between colorectal cancers with or without TP53 inactivating mutations (left), microsatellite instability (middle) and genomic instability (chromosomal instability; right). Sample numbers are shown in the parentheses. P values from two-sided t-test (left, middle) and linear regression (right) were shown. Box plots illustrate median values with interquartile ranges (IQR) with whiskers (1.5 x IQRs). Blue lines represent the regression lines, and the shaded areas indicate their 95% confidence intervals. P values from two-sided multivariate regression were represented in the right space. ns, not significant. d. In 19 matched colorectal cancer tissues. e. In 19 matched colorectal cancer tissues and 52 PCAWG colorectal cancer tissues. f. In 19 matched colorecal cancer tissues and 4 cancer types (colorectal adenocarcinomas, oesophageal adenocarcinomas, lung squamous cell carcinomas, and head and neck squamous cell carcinomas) showing a higher soL1R burden among 40 histologic types in PCAWG. g. In 19 matched colorectal cancer tissues with all whitelist PCAWG samples.

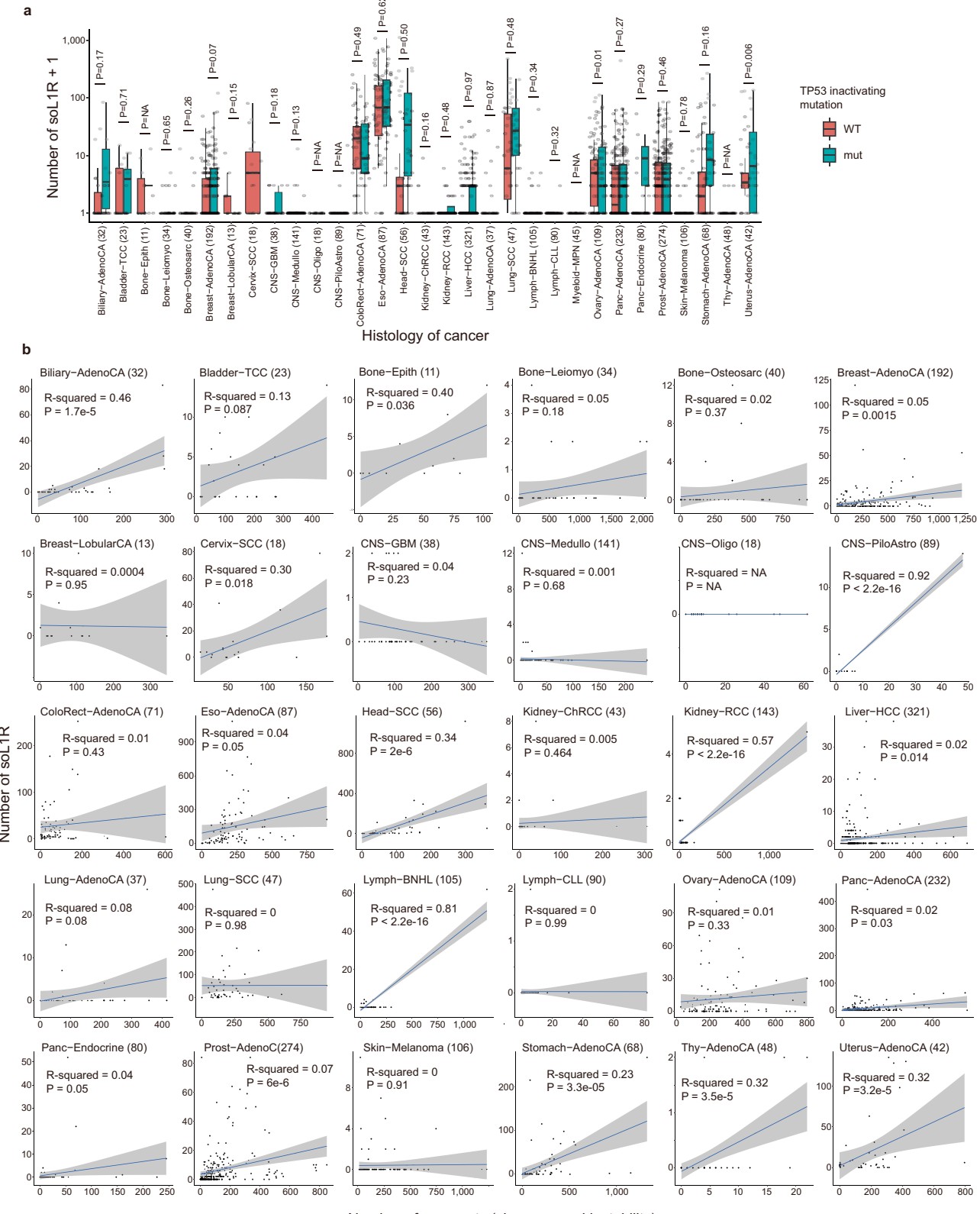

**Extended Data Fig. 11 | SoL1Rs in cancer and classical genome instability.**
Relationship between soL1R burden and classical genome instability, such as
TP53-inactivating mutations and chromosomal instability, was analysed in
PCAWG whitelist samples (n = 2,677) and 19 matched colorectal cancers in this
study. Cancer types with less than 10 cases were not considered. **a**, Somatic
TP53-inactivating mutations and the number of soL1R events. Box plots
illustrate median values with interquartile ranges (IQR) with whiskers (1.5 x

IQRs). Number of cases in each histology type were shown in parentheses. P values
from two-sided t-test were shown. NA, not available. **b**, Linear regression
between the chromosomal instability (genomic rearrangements) and the
number of soL1R events in each cancer type. Blue lines represent the regression
lines, and the shaded areas indicate their 95% confidence intervals. R-squared
and P values from linear regression were represented in each panel.

# Reporting Summary

## Statistics

For all statistical analyses, confirm that the following items are present in the figure legend, table legend, main text, or Methods section.

| n/a | Confirmed | |
|---|---|---|
| ☐ | ☒ | The exact sample size (*n*) for each experimental group/condition, given as a discrete number and unit of measurement |
| ☐ | ☒ | A statement on whether measurements were taken from distinct samples or whether the same sample was measured repeatedly |
| ☐ | ☒ | The statistical test(s) used AND whether they are one- or two-sided *Only common tests should be described solely by name; describe more complex techniques in the Methods section.* |
| ☐ | ☒ | A description of all covariates tested |
| ☐ | ☒ | A description of any assumptions or corrections, such as tests of normality and adjustment for multiple comparisons |
| ☐ | ☒ | A full description of the statistical parameters including central tendency (e.g. means) or other basic estimates (e.g. regression coefficient) AND variation (e.g. standard deviation) or associated estimates of uncertainty (e.g. confidence intervals) |
| ☐ | ☒ | For null hypothesis testing, the test statistic (e.g. *F*, *t*, *r*) with confidence intervals, effect sizes, degrees of freedom and *P* value noted *Give P values as exact values whenever suitable.* |
| ☒ | ☐ | For Bayesian analysis, information on the choice of priors and Markov chain Monte Carlo settings |
| ☒ | ☐ | For hierarchical and complex designs, identification of the appropriate level for tests and full reporting of outcomes |
| ☒ | ☐ | Estimates of effect sizes (e.g. Cohen's *d*, Pearson's *r*), indicating how they were calculated |

*Our web collection on statistics for biologists contains articles on many of the points above.*

## Software and code

Policy information about availability of computer code

| Data collection | No software was used. |
|---|---|
| Data analysis | We aligned whole-genome sequencing reads to the human reference genome (GRCh37) using BWA(0.7.17) algorithm. The duplicated reads were removed by Picard (2.1.0) or SAMBLASTER(0.1.24). We identified single-nucleotide variants and short indels using Varscan2(2.4.2) and HaplotyperCaller2 in GATK(4.0.0.0). In addition, we identified somatic genomic rearrangements using DELLY(0.7.6) and called somatic L1 retrotransposition using MELT(2.2.0), TraFiC-mem(1.2.0), xTea(0.1), as well as DELLY(0.7.6). Detected variants were inspected using IGV(2.8.2). Long-read sequences generated by the PacBio platform were aligned to human reference genome (GRCh37) using pbmm2(1.4.0). We removed the adaptor sequences from RNAseq and EM-seq reads using Cutadapt(1.18) and aligned to the human reference genome (GRCh37) using BWA(0.7.17) and Bismark(0.22.3), respectively. Signature analysis was performed using SigProfiler(1.1.4) and HDP(0.1.5). Custom scripts were written by Python(3.6.10, 2.7) and R(3.6.0) and are available at Github (https://github.com/ju-lab/colon_LINE1). |

For manuscripts utilizing custom algorithms or software that are central to the research but not yet described in published literature, software must be made available to editors and reviewers. We strongly encourage code deposition in a community repository (e.g. GitHub). See the Nature Portfolio guidelines for submitting code & software for further information.

## Data

Policy information about availability of data

All manuscripts must include a data availability statement. This statement should provide the following information, where applicable:
- Accession codes, unique identifiers, or web links for publicly available datasets
- A description of any restrictions on data availability
- For clinical datasets or third party data, please ensure that the statement adheres to our policy

> Whole-genome, DNA methylation, and transcriptome sequencing data are deposited in the European Genome-phenome Archive (EGA) with accession EGAS00001006213 and available for general research use. Human reference genome (GRCh37) is available at NIH websites (https://www.ncbi.nlm.nih.gov/data-hub/genome/GCF_000001405.13).

## Human research participants

Policy information about studies involving human research participants and Sex and Gender in Research.

| | |
|---|---|
| Reporting on sex and gender | Sex information of the study participants were initially provided from hospital and confirmed by sequencing depth of sex chromosomes in whole-genome sequencing. The information is available in Supp Table 1. We did not find any differences in L1 activity between males and females. Sex and gender were not considered in study design. |
| Population characteristics | Out of 28 patients, 20 were diagnosed with colorectal cancer. The age of the patients spanned several age groups ranging from 37 to 93. The ratio between males and famales were almost 1:1 (13 males and 15 females). |
| Recruitment | We recruited participants from those who planned to undergo colectomy surgery (n=19) or colonoscopic colon polypectomy (n=1) in Seoul National University Hospital. Participants were identified through a review of medical records as were approached in the clinic to obtain informed consent. There is a potential bias since all recruited participants had a diagnosis of colorectal disease. Data for the other individuals (7 for fibroblast and 1 for blood clones) were published previously and downloaded for this study. |
| Ethics oversight | All the procedures in this study were approved by the Institutional Review Board of Seoul National University Hospital (approval number: 1911-106-1080) and Korea Advanced Institute of Science and Technology (approval number: KH2022-058). |

Note that full information on the approval of the study protocol must also be provided in the manuscript.

# Field-specific reporting

Please select the one below that is the best fit for your research. If you are not sure, read the appropriate sections before making your selection.

☒ Life sciences      ☐ Behavioural & social sciences      ☐ Ecological, evolutionary & environmental sciences

For a reference copy of the document with all sections, see nature.com/documents/nr-reporting-summary-flat.pdf

# Life sciences study design

All studies must disclose on these points even when the disclosure is negative.

| | |
|---|---|
| Sample size | No statistical methods were used to predetermine sample size. We selected samples from available individuals to describe the mutational landscape of LINE1 retrotransposition in normal cells. |
| Data exclusions | No data were excluded from the analyses. |
| Replication | We repeated the colon organoid culture and generated 13 pairs of mother-daughter colon organoids. The main purpose was to estimate the rate of culture-associated L1s, but all the somatic L1 retrotransposition events identified in mother organoids were validated in daughter organoids. |
| Randomization | We did not perform randomization because this study did not involve experimental groups. Covariates were controlled by statistical methods. |
| Blinding | Blinding is not applicable because this is a descriptive study. |

# Reporting for specific materials, systems and methods

We require information from authors about some types of materials, experimental systems and methods used in many studies. Here, indicate whether each material, system or method listed is relevant to your study. If you are not sure if a list item applies to your research, read the appropriate section before selecting a response.

## Materials & experimental systems

| n/a | Involved in the study |
|-----|------------------------|
| ☒ | Antibodies |
| ☒ | Eukaryotic cell lines |
| ☒ | Palaeontology and archaeology |
| ☒ | Animals and other organisms |
| ☒ | Clinical data |
| ☒ | Dual use research of concern |

## Methods

| n/a | Involved in the study |
|-----|------------------------|
| ☒ | ChIP-seq |
| ☒ | Flow cytometry |
| ☒ | MRI-based neuroimaging |

