## [Peer Review File · Nature]

Manuscript Title: Widespread somatic L1 retrotransposition in normal colorectal epithelium

Reviewer Comments & Author Rebuttals

Reviewer Reports on the Initial Version:

Referee #1: retrotransposons in cancer

Referee #2: cancer evolution, genomics

Referee #3: cancer evolution, genomics

Referees' comments:

Referee #1 (Remarks to the Author):

Ju's team has analysed the patterns and mechanisms of L1 somatic integration in normal and diseased tissues. They have obtained clonal cell lines from individual cells isolated from human tissues, and performed multiomics, including genome, transcriptome and methylome sequencing. In total they have analysed 892 single-cell clones (mainly from colon) from 28 individuals. They make a description of the L1 somatic landscape between tissues and along development. They find somatic L1s occurring early in embryogenesis, which is in correlation with L1 demethylation of source elements, and the majority of L1 transcripts do not generate L1 somatic insertions.

The general aim of the study is very attractive and interesting for the field of somatic retrotransposition, because we do not know too much about retrotransposition in normal tissues, and we also do not know about subclonal retrotransposition in tumours. Some results are interesting and novel. The authors find, for instance, that retrotransposition rate is higher in embryogenesis than in somatic lineages after birth! However, the main problem of this manuscript is the lack of novelty. Many of the findings, although interesting, recapitulate what has been already described in cancer and so they do not represent novel insights into the biology of transposons. The part describing the landscape of somatic L1 in normal tissues relative to development is amazing. Combining genomics with expression and methylation data is novel. However, many conclusions in this work are obvious and do not provide too much new insights to the field or the results are not surprising. For instance, retrotransposition burden is higher in tumours relative to normal tissues (line 179); other optional events can be engaged for retrotransposition (i.e., twin priming; line 272); different rates of activity found in source elements (line 306); L1 promoter methylation shows strong negative correlation with RNA expression (line 351). Some other conclusions may require further review of the transposon literature: for the claiming of novel source elements (line 283) the authors should check supplementary tables S22 and S23 from PMID 33632895; for the conclusion that ultra-rare source elements are likely to be acquired recently in the germline (line 327), see reference 33632895; for the conclusion that promoter demethylation of hot L1s is necessary for somatic retrotransposition, see for instance reference PMID 25082706. Why the authors have not identified

other retrotransposition events (Alu, SVA, ERV)?

I wonder why the authors have not focused more on the embryogenesis. The conclusions in that part of the manuscript are ground-breaking and findings are relevant. The finding of correlation between retrotransposition and age (line 143) is very relevant, but it is not exploited sufficiently in the manuscript. The association of somatic retrotranspositions and the number of cell divisions is amazing. Finally, the combination of methylation and transcriptomic data with the genomics, provide very unique results; one of the best sections of the paper are the last two paragraphs (lines 372-394). Again, the authors do not exploit this enough. These conclusions should be the central part of the manuscript, but they are not...

The methods (molecular biology and computational) employed are cutting-edge, although there is not much innovation. For instance, identification of L1 somatic retrotranspositions is based on a method described in 2014, which can be improved to identify more source elements. The authors are assuming a fixed rate of mutation as a molecular clock, from reference 8, that is later used to perform very precise timing (i.e., cell divisions) of somatic retrotranspositions. I really love it, but I wonder how precise this is...

Referee #2 (Remarks to the Author):

This is an interesting and clearly presented manuscript describing the somatic evolution of somatic line-1 retrotransposition (sol1R) in ageing human cells. By sequencing single cell-derived colonies from patient material, and combining these genomic data with gene expression and DNA methylation data, the authors report the prevalence, rate and clonality of sol1Rs, finding evidence of increased rate of sol1R in embryogenesis and tumorigenesis. There are interesting results that utilise human population scale data that suggest sol1R is generally under negative selection (with implications for understanding how this negative selection is relaxed in carcinogenesis). The final section of the manuscript combines multi-omic data to explore potential molecular regulators of sol1R, finding (perhaps unsurprisingly) that sol1R requires the L1 to be unmethylated and expressed in order to be inserted; this final section is the least developed of the paper. Overall the manuscript offers careful and quite thorough insights into the somatic evolution of L1 elements in normal human tissues.

Comments:

1. I think the sol1R detection in cancers is on bulk tissue (where there are likely many genetically diverged lineages, that each could have a different sol1R) so the comparison of sol1R burden with normal clones should be tempered, or alternatively, a comparison of sol1R burden where cancer cell fraction (CCF) is estimated to be 1 could be compared. This is an important analysis as it supports/refutes the claim that sol1R burden is increased in cancer.

2. What is the interrelationship between copy number alterations (CNAs) and other structural

alterations (SAs) and sol1Rs? Could it be that some sol1R events are actually duplications caused by other SAs? (are these the complex events referred to in Figure 2B?) I would think the analysis is robust to this (as the authors examine flanking regions of L1 insertion sites) but it would be good to confirm. Relatedly, do the authors think that the purported higher burden of sol1Rs in colorectal adenomas/cancers is interrelated with tolerance of chromosomal instability that is common in these cancers? (the authors could explicitly assess sol1R burden in MSI vs MSS cases, and/or genome doubled cancers, and repeat the regression in Fig 2d comparing normal vs cancer, perhaps making use of PCAWG cases)

3. Are there mutational consequences of sol1R insertion at the insertion points (analogous to kaetags)?

4. Related to Figure 2, do features of insertion sites differ through molecular time?

5. Regarding the multi-omic analysis of sol1Rs, is there evidence that there are genome-wide differences in DNA methylation (and/or expression) in clones with sol1Rs (perhaps further evidence of a “genomic instability tolerant” phenotype)?

Similarly, is there evidence of changed methylation status (and so changed chromatin status) at insertion sites? (and the authors could speculate on cause/consequence) The analysis as-is is restricted to looking at the molecular features of sol1R sites only; it feels that more interesting analysis could come from these multi-omic data from a more extensive analysis.

6. (minor) The statement that “Third, we observed a high level of cell-type specificity in the sol1R burden.” as evidence that sol1R isn’t culture related seems weak: presumably culture conditions were different between cell types, and cell types differentially adapt to culture, which could conceivably cause sol1R.

7. (minor) All the adenomas appear to be from one individual with a germline-caused (MUTYH) mutator phenotype, so these adenomas are unlikely to represent the wider population – this should be acknowledged.

Referee #3 (Remarks to the Author):

Nam and Youk et al present a detailed study on L1 retrotransposition during development from a large number of expanded cellular colonies from healthy adults. Overall the study is detailed and well written, with a really nice narrative throughout the paper. The paper also does a nice job complementing other recent works exploring human lineage reconstruction with a variety of mutational patterns (SNV, WGS+sanger, etc), putting this work into perspective (Evrony, etc). The characterization of new hot L1s and mechanisms is a really nice angle, and the detail throughout the paper is appreciated. The figures are also really well done and visually explain the story well. My major concerns are about the generalizability from the tissues sampled here to the more global view of L1 retrotransposition.

Specific feedback:

One big concern is the disconnect between the promise of L1 retrotransposon characterization in normal tissue vs the limited number of tissues tackled here. The focus is really the colon; one option would be to tone back the language a bit. For instance you say “In this study, we observed a substantial level of somatic mosaicism in normal cells driven by soL1Rs. However, many things are yet to be discovered. For example, some other cell types not investigated in this study may have higher soL1R rates than colorectal epithelium”, but many points in the paper speak to the global perspective (“cell types not in this study” is almost all cell types). Certainly the ideal option would be to validate with PCR (or other targeted approach) some of the L1 transpositions you see here in other tissue samples from the same patient, especially to validate the lineage information. But more clearly stated limitations in the discussion would be helpful.

At line 179 (thank for line numbers!) you say that tumor cells have rates about 10X higher than normal cells. In figure 1g/h you also detail the huge increase in mutations in the carcinomas, could you detail the L1 rate per inferred molecular time / cell division for the tumors? Sorry if this doesn't make sense, but I'm curious if the increased L1 'rate' is a by-product of the increased cell division count. This goes to the closing point at line 394 as well, as this could affect the conclusion about defense mechanisms (?)

The double integration is a nice catch! (277)

If you include the four dominant hot L1s in the ultra-rare vs common L1s, I assume this changes the conclusion? It seems a bit strange to exclude these, but maybe this is getting to the next point.

You have the mapping locations for the 12p13.32 hot L1, which has nearly complete methylation of the promoter across the clones. Could you fill in more details when you speculate “have crucial functional roles or are located in genomic loci that are difficult to repress.” (331)? This seems important given the detail you put into methylation for some of the other L1s. I worry some people will see the methylation story as very dependent on the individual L1, and it would be nice to have a more detailed story about why these escape the typical pattern (again to the points made around ~381).

Author Rebuttals to Initial Comments:

Referee #1 (Remarks to the Author): expertise-retrotransposons in cancer

Comment 1-1. Ju's team has analysed the patterns and mechanisms of L1 somatic integration in normal and diseased tissues. They have obtained clonal cell lines from individual cells isolated from human tissues, and performed multiomics, including genome, transcriptome and methylome sequencing. In total they have analysed 892 single-cell clones (mainly from colon) from 28 individuals. They make a description of the L1 somatic landscape between tissues and along development. They find somatic L1s occurring early in embryogenesis, which is in correlation with L1 demethylation of source elements, and the majority of L1 transcripts do not generate L1 somatic insertions.

Answer 1-1

We appreciate the reviewer's time in reviewing our manuscript and the valuable and constructive comments which make our work more meaningful. It is an excellent summary of our work.

Comment 1-2. The general aim of the study is very attractive and interesting for the field of somatic retrotransposition, because we do not know too much about retrotransposition in normal tissues, and we also do not know about subclonal retrotransposition in tumours. Some results are interesting and novel. The authors find, for instance, that retrotransposition rate is higher in embryogenesis than in somatic lineages after birth!

Answer 1-2

The reviewer describes one of the main findings of our work. To abide by the reviewers' points, we have conducted more sophisticated analyses on the embryogenesis part, and the conclusions of our revised manuscript have become more robust.

Comment 1-3. However, the main problem of this manuscript is the lack of novelty. Many of the findings, although interesting, recapitulate what has been already described in cancer and so they do not represent novel insights into the biology of transposons. The part describing the landscape of somatic L1 in normal tissues relative to development is amazing. Combining genomics with expression and methylation data is novel. However, many conclusions in this work are obvious and do not provide too much new insights to the field or the results are not surprising. For instance, **1-a)** retrotransposition burden is higher in tumours relative to normal tissues (line 179); **1-b)** other optional events can be engaged for retrotransposition (i.e., twin priming; line 272); **1-c)** different rates of activity found in source

elements (line 306); **1-d**) L1 promoter methylation shows strong negative correlation with RNA expression (line 351).

Answer 1-3

We appreciate the reviewer's comments and criticisms. The reviewer mentions that many of our descriptions do not provide further insights. Some of our observations can be speculated from extrapolation from what was observed in cancer tissues. For example, issues c and d were described in cancers (Tubio et al., Science 2014), and twin priming (issue b) was previously proposed (Ostertag and Kazazian., Genome Res. 2001).

Although we respect the reviewer's criticisms and understand what the reviewer means, we carefully disagree with the reviewer's opinion that our manuscript lacks novelty. These issues are NOT our "final conclusions" but "intermediate findings," which should be robustly confirmed for our final conclusion. To our best knowledge, this is the first work that reveals the detailed landscape of somatic L1 retrotranspositions (soL1Rs) in human "non-neoplastic" cells at the absolute single-cell resolution. Therefore, the confirmation of previous speculations should be conducted for downstream analyses. In addition, we believe that our work illustrates the characteristics of soL1Rs in more detail.

-Issue 1-a) We identified that soL1R rates differ across normal cell lineages at different developmental timing, for the first time (revised Fig. 1i).

[Revised manuscript, page 7, Results, Section "High soL1R activity in embryogenesis"]

To compare soL1R rates in various stages and cell types, we normalised the soL1Rs using endogenous point mutation (SBSs 1 and 5). The soL1R rate in the colorectal epithelial lineages after birth was 1.2 soL1Rs per 1,000 endogenous point mutations. The rate was ~4 times higher in the post-gastrulation embryonic stages differentiating to the colorectal lineages (4.52 per 1,000 endogenous point mutations; $P=8.6 \times 10^{-4}$, two-sided Poisson exact test), which is equivalent to $1.1 \times 10^{-3} - 9.0 \times 10^{-3}$ soL1R per cell per early cell division (assuming the fixed early endogenous point mutation rate^{8,34}). Pre-gastrulation stage embryos showed 1.06 soL1Rs per 1,000 endogenous point mutations, although we observed only one instance from the 27 individuals. In contrast, the rates were close to 0 soL1Rs per 1,000 endogenous point mutations for the blood and fibroblast lineages, regardless of embryonic and post-embryonic stages (**Fig. 1i**).

Fig. 1i (new). Normalised soL1R rates, estimated by the number of soL1Rs per 1,000 somatic point mutations, in various stages and cell types.

In addition, we identified how high the soL1R rate is in the tumour cells. In soL1R burden (per cell), a colorectal tumour tissue (the most recent common ancestral cell of a tumour tissue) harbors ~10 times more soL1Rs compared to its matched normal colorectal epithelial cell. In terms of soL1R rate, a colorectal tumour tissue showed a 3-times higher soL1R rate per endogenous point mutations (molecular clock). These two findings were not extensively described previously, to our best knowledge.

[Revised manuscript, page 12, Results, Section “Acceleration of soL1R activity during tumourigenesis”]

Lastly, we sought to determine the soL1R burden and rate in tumourigenesis. All 19 matched colorectal carcinomas harboured soL1R events with considerable variance ranging between 4 and 105 (**Fig. 1b**). Most soL1R events identified in cancer were clonal ($n=556$; 97.2%). On average, the number of clonal soL1R events was 29 per cancer, approximately 10 times more frequent than observed in normal colorectal clones. The soL1R rate in colorectal carcinomas was 3.47 per 1,000 endogenous point mutations, which is ~3 times higher than estimated in normal colorectal lineages (**Extended Data Fig. 12a**).

Extended Data Fig. 12a (new). The soL1R rate is accelerated during tumourigenesis in colorectal lineages.

-Issue 1-b) The ‘twin priming’ is just an observation, not a final conclusion of this study. We observed twin priming events in 29.6% of soL1Rs in the normal colorectal epithelium and described it in the manuscript. Although the proportion is just a number, we carefully claim that it was not known in the normal (colorectal) cells. In addition, the short foldback inversion (inverted duplications) in the 5’ upstream of the target site has not been described previously.

[Revised manuscript, page 11, Results, Section “Mechanisms of soL1Rs inferred from the breakpoints”]

We further investigated breakpoint sequences at the *soL1R* target sites to infer the mechanistic processes of L1 insertions. In addition to the two canonical features (TSD and poly-A tail), which are acquired by target-primed reverse transcription (process A; **Figs. 4g, 4h**), a substantial fraction of *soL1Rs* showed variations in the head sequence of the inserts, characterised by (1) short inversion in the intra-RT body ($n=356$; 29.6%), (2) short foldback inversion (inverted duplications) in the 5' upstream of the target site ($n=3$; 0.2%), or (3) both ($n=1$; 0.1%). These sequence variations could be explained by the twin priming mechanism (process B; **Fig. 4h**)⁵⁰ and additional DNA synthesis (52~220 bp) potentially by DNA polymerases in the final resolution of the L1-mediated insertional mutagenesis (process C; **Fig. 4h**). The latter mechanism has not been described previously, to our best knowledge.

Figs. 4g, 4h (previously **Figs. 2e, 2f**). Schematic diagrams of genomic structures of canonical and complex L1 insertions (**g**) and underlying mechanisms (**h**). RT body, retrotransposed body; TSD, target site duplication; DSB, double-strand break.

-Issue 1-c) The retrotransposition activity of different source elements was comprehensively measured in the normal colorectal epithelium in this study. Although the differential activity of source elements was reported in cancer (Rodriguez-Martin et al., *Nature Genetics*, 2020), it was in cancers, not in normal cells. We carefully estimated the L1 rates in normal cells, with the number of transduction, L1 alleles, and endogenous point mutations, for the first time to our knowledge. Again, it is not our 'final conclusion'. We observed the 'inverse correlation' between the population allele frequency of L1 sources and their retrotransposition activity in normal colon tissues (except for the four 'prevalent-active' L1 sources; **Fig. 2c** in the revised manuscript).

Fig. 2c (previously 3b). Relationship between the population allele frequency of rc-L1 and their normalised retrotransposition activity. Green dots indicate private sources present in only one individual in our colorectal cohort. Red dots indicate common sources, but showing higher retrotransposition activity than expected. Gray dots indicate common sources with no transduction events in our study. TPAM, the number of transductions per allele per 1 million clock-like mutations of molecular time.

Further, to understand the foundation of differential retrotransposition activity across source elements, we conducted additional whole-genome DNA methylation sequencing (n=139 in the revised manuscript; previously n=29 in the original manuscript) and long-read whole-genome sequencing (2 clones). Overall, prevalent sources in the human population were old L1s (PA2, preTa; new **Fig. 2d**), frequently harbored ORF-truncating mutations, and/or promoter demethylation (new **Fig. 3e**).

Fig. 2d (new). Proportion of sources in each L1 subfamily and sources with truncating mutations across different PAF groups. PAF, population allele frequency.

Fig. 3e (new). Proportion of non-truncated alleles and clones with demethylated promoters in 89 population-prevalent rc-L1s. Red dots indicate four prevalent-active sources. Black and gray dots indicate common sources with any and no transduction events in our study, respectively.

-Issue 1-d) For the downstream analyses, we needed to confirm the strong negative correlation between L1 promoter methylation and L1 transcription at the single-clone level. Otherwise, our promoter methylation analyses (main Figure 3 in the revised manuscript) would be less meaningful because it indicates that L1 promoter methylation does not affect L1 activity. As the reviewer can also feel, the negative correlation is not our final conclusion but is described as a prerequisite condition that should be confirmed before the methylation landscape analyses. The negative correlation is just simply mentioned in the revised manuscript.

[Revised manuscript, page 9, Results, Section “Promoter demethylation of rc-L1s”]

To explore the epigenetic foundation of the differential rc-L1 activities in normal cells, we conducted multi-omics analyses by integrating whole-genome DNA methylation (139 clones), RNA expression profiles (116 clones), and early clonal phylogenies of the clones in 9 individuals (**Figs. 1a, 3a**). As reported in bulk tissues^{18,41}, a strong negative correlation between L1 promoter demethylation and transcription from the source elements was observed in the clones (**Extended Data Fig. 6a**).

Comment 1-4. Some other conclusions may require further review of the transposon literature: **1-e)** for the claiming of novel source elements (line 283) the authors should check supplementary tables S22 and S23 from PMID 33632895; **1-f)** for the conclusion that ultra-rare source elements are likely to be acquired recently in the germline (line 327), see reference 33632895; **1-g)** for the conclusion that promoter demethylation of hot L1s is necessary for somatic retrotransposition, see for instance reference PMID 25082706. **1-h)** Why the authors have not identified other retrotransposition events (Alu, SVA, ERV)?

Answer 1-4

We appreciate the reviewer's comment as these are constructive.

-Issue 1-e) We checked the paper (Ebert et al., Science 2021; citation #13 in the revised manuscript) and other papers, and now we have 264 previously known sources that can be retrotransposed (which was 124 in the original manuscript). The number of novel rc-L1s has slightly decreased (now 12, previously 15), and we have updated our novel source elements accordingly. We cited the paper in the section of the timing of ultra-rare L1 insertion as well. We appreciate the reviewer for the comment.

[Original manuscript, #86-#89, "Introduction"]

However, most of the approximately 500,000 L1s in the human reference genome are now molecular fossils, or unable to transpose further, because they are truncated and have lost their functional potential. Approximately 120 L1s are known as retrotransposition-competent, called "hot L1s"^{15,16}.

⇒

[Revised manuscript, page 3, "Introduction"]

However, most of the approximately 500,000 L1s in the human reference genome are now molecular fossils or unable to transpose further because they are truncated and have lost their functional potential. Of these, the retrotransposition competency of 264 L1s have been shown in cancers¹⁷⁻¹⁹ and/or in other experimental conditions^{13,14,20-22}.

[Original manuscript, #284, Results, Section "Hot L1 polymorphism across populations"]

From the 241 transduction events detected in clones and cancer tissues, 34 hot L1 sources were identified (Supplementary Table 3). Of these, 15 (44%) were new and did not overlap with the 124 previously known sources¹⁵.

⇒

[Revised manuscript, page 7, Results, Section "Tracing the source element of soL1Rs"]

*For 241 transduction events detected from clones and cancer tissues in this study, 34 L1s were responsible sources (**Fig. 2b; Supplementary Table 4**), which were confirmed as retrotransposition-competent L1s (**rc-L1s**). Of these, 12 (35%) did not overlap with previously known retrotransposition-competent sources ($n=264$)^{13,14,17-22}.*

-Issue 1-f) We cited the paper in the revised manuscript. In addition, we also explored the subfamilies of the population-frequent L1 sources using long-read genome sequencing. The subfamily information is represented in Fig. 2d and Supplementary Table 4 of the revised manuscript.

[Original manuscript, #299, Results, Section "Hot L1 polymorphism across populations"]

Of note, four hot L1s (17q25.3, 1q23.3-1, 2q21.1, and 1p22.1) were private to an individual in our cohort but not observed in the population panel, indicating that these are ultra-rare sources that are likely to be acquired recently in the germline (Fig. 3a).

⇒

[Revised manuscript, page 8, Results, Section “Differential soL1R activity across source elements”]

Ultra-rare sources may maintain their high activity because they emerged relatively recently¹³, prior to sufficient negative regulation.

To understand the genetic foundation of the differential activities across rc-L1s, we explored sequence polymorphisms of the source elements using long-read whole-genome sequencing of two colorectal clones. Population-prevalent source elements were predominantly in the older L1 subfamilies (such as pre-Ta and PA2), as suggested previously¹³, and harboured ORF-disrupting mutations more frequently than rare source elements (Fig. 2d; Supplementary Table 4).

-Issue 1-g We minimized and updated the section and explicitly cited the paper (Tubio et al., Science 2014; citation #21 and #18 in the previous and revised manuscript, respectively) in the related sentence. Just for information, the last author of the current manuscript (Young Seok Ju) is the co-second author of the paper.

[Original manuscript, #342, Results, Section “Epigenetic regulation of hot L1 activation”]

Of note, DNA methylation of the hot L1 promoter has been accepted as a key mechanism for inhibiting L1 transcriptional activation^{21,39}. Transcription of each hot L1 source can be specifically assessed by its read-through transcripts in the 3' downstream region⁴⁰. The profiles of promoter methylation and RNA expression varied across hot L1s (Figs. 4b-f; Extended Data Fig. 7). For instance, a hot L1 12q13.13 showed full demethylation for both alleles in most clones, and RNA transcription was generally observed in the 3' downstream region (Fig. 4b). In contrast, a hot L1 9q32 represented overall methylation and silent gene expression, particularly in clones from HC15 (Fig. 4c). A hot L1 12p13.32 displayed a mixed methylation pattern across clones, predominantly with full methylation and full demethylation (Fig. 4d). The L1 promoter methylation profiles showed a strong negative correlation with the RNA expression levels (Figs. 4e, f)

⇒

[Revised manuscript, page 9, Results, Section “Promoter demethylation of rc-L1s”]

As reported in bulk tissues^{18,41}, a strong negative correlation between L1 promoter demethylation and transcription from the source elements was observed in the clones (Extended Data Fig. 6a).

-Issue 1-h) We conducted additional analyses for Alu and other mobile elements. It was neglected in the previous manuscript, as their activity should be much lower than that of L1. We finally identified 9 somatic Alu insertion events in normal clones, which is less than 1% of the burden of soL1Rs. We did not find somatic mobilization of other types of transposons. Interestingly, one somatic Alu insertion was an embryonic event, which is briefly introduced in the revised manuscript and Extended Data Fig. 4. We appreciate the reviewer's suggestion which made our manuscript more meaningful.

[Revised manuscript, page 5, Results, Section "Widespread soL1R in normal colorectal epithelium"]

For the other retrotransposon types, we additionally detected 9 somatic Alu insertions in the normal clones (**Supplementary Table 2**).

[Revised manuscript, page 6, Results, Section "High soL1R activity in embryogenesis"]

Similarly, one somatic Alu insertion, found in HC04, was also likely obtained in the pre-gastrulation stage (**Extended Data Fig. 4**).

Part of Extended Data Fig. 4 (modified). An early embryonic Alu insertion event identified in HC04.

Comment 1-5. I wonder why the authors have not focused more on the embryogenesis. The conclusions in that part of the manuscript are ground-breaking and findings are relevant. **1-i)** The finding of correlation between retrotransposition and age (line 143) is very relevant, but it is not exploited sufficiently in the manuscript. **1-j)** The association of somatic retrotranspositions and the number of cell divisions is amazing. Finally, **1-k)** the combination of methylation and transcriptomic data with the genomics, provide very unique results; one of

the best sections of the paper are the last two paragraphs (lines 372-394). Again, the authors do not exploit this enough. These conclusions should be the central part of the manuscript, but they are not...

Answer 1-5.

We appreciate the reviewer for this comment. To abide by the reviewer's comment, the revised manuscript focuses on the embryogenesis part more extensively. To this end,

1. We added whole-genome methylation sequencing from 103 additional clones. Multiomics analyses are now conducted using 132 clones (which was 29 in the original manuscript).
2. To highlight the embryogenesis part, we moved these descriptions to earlier sections. For example, in the original manuscript, soL1Rs in cancer samples and mechanisms were shown in Figure 2, and the embryogenesis parts were split in the early and late parts of the manuscript (Figure 1 and Figure 4). Now, the cancer/mechanism part is moved to Figure 4, and the embryogenesis part is consistently described from Figures 1-3.
3. To highlight the embryogenesis part more efficiently, we subdivided the embryogenic periods into the 'pre-gastrulation stage', 'post-gastrulation stage, differentiating into colorectal epithelium', 'post-gastrulation stage, differentiating into blood/fibroblasts'. We added 7 whole-genome methylation sequencing from 7 fibroblast clones.

With these efforts, the revised manuscript highlights the embryogenesis part more thoroughly and becomes more meaningful than the original manuscript. Below are our answers for each individual topic.

-Issue 1-i) The section has been rephrased to highlight the age correlation more efficiently. The corresponding figure (Fig. 1e) was updated with inter-clonal variation in an individual.

[Original manuscript, #150, Results, Section "SoL1R rates in normal colorectal epithelium"]

However, there were substantial variations in soL1R burden within and between individuals. For example, in HC15, the soL1R burden ranged between 1-18 across the 23 clones of the individual (Fig. 1d). Overall, 184 soL1Rs were identified from HC15 (8 soL1Rs per clone), which was a 2.6-fold higher number than random expectation ($P=9.7\times 10^{-30}$, two-sided exact Poisson test). For each individual, the average soL1Rs rates showed a positive relationship with age, with approximately 0.028 soL1Rs per clone per year (Fig. 1e), similar to the clock-like property of endogenous somatic SNVs and indels (Extended Data Fig. 2a)³⁰.

⇒

[Revised manuscript, page 5, Results, Section "Widespread soL1R in normal colorectal epithelium"]

*In colorectal epithelium, we found substantial variations in soL1R burden across clones and individuals. The soL1R burden in colorectal clones was 0–18 per clone (**Fig. 1d**). When averaged, soL1R burdens showed a positive relationship with the age of individuals (0.028 soL1Rs per clone per year; **Fig. 1e**), similar to the clock-like property of endogenous somatic*

SNVs and indels (**Extended Data Fig. 2a**)³⁷. It implies that soL1Rs in the colorectal epithelium are acquired at a more-or-less constant background rate throughout life. However, we found two outlier individuals (**Fig. 1e**), which suggests potential genetic predisposition and/or some environmental exposures that stimulate L1 activities. For instance, clones of HC15 showed a 2.6-fold higher burden on average than randomly expected ($P=9.7\times 10^{-30}$, two-sided exact Poisson test).

Fig. 1e (modified). Linear regression of the average number of soL1Rs per clone on age. Vertical line indicates the range of soL1R burden per clone in individuals. Blue line represents the regression line, and shaded areas indicate a 95% confidence interval of the regression line. Two outlier individuals (HC15 and HC06) are highlighted in red.

-Issue 1-j) We appreciate the reviewer for the comment. We subdivided the soL1R rates (per endogenous point mutations) into the pre-gastrulation stage, post-gastrulation embryonic stages (for three cell types including colorectal epithelium, blood, and fibroblast lineages), and post-embryonic stages (for the three cell types). These rates are shown in new Fig. 1i, and the updated paragraph is shown below.

[Original manuscript, #210, Results, Section “L1 retrotransposition starts during embryogenesis”]

Of 19 phylogenies by normal colorectal clones, informative embryonic lineages (early shared branches) included a total of 2,827 mutations of molecular time (Extended Data Figs. 3, 4). Therefore, our observation (10 embryonic soL1Rs) suggests one soL1R per 283 somatic point mutations in the early stage of embryonic development (95% CI: one in every 175-744 mutations; equivalent to $1.6\times 10^{-3} \sim 6.8\times 10^{-3}$ soL1R per cell per cell division), which was 3.3-fold higher than the rate at the late somatic lineages unique to single clones (one per 940 clock-like mutations in molecular time; $P=0.001$, two-sided Poisson exact test). Our findings indicated that the soL1R rate is higher in embryogenesis than in somatic lineages after birth.

⇒

[Revised manuscript, page 7, Results, Section “High soL1R activity in embryogenesis”]

To compare soL1R rates in various stages and cell types, we normalised the soL1Rs using endogenous point mutation (SBSs 1 and 5). The soL1R rate in the colorectal epithelial lineages after birth was 1.2 soL1Rs per 1,000 endogenous point mutations. The rate was ~4 times higher in the post-gastrulation embryonic stages differentiating to the colorectal lineages (4.52 per 1,000 endogenous point mutations; $P=8.6\times 10^{-4}$, two-sided Poisson exact test), which is equivalent to $1.1\times 10^{-3} - 9.0\times 10^{-3}$ soL1R per cell per early cell division (assuming the fixed early endogenous point mutation rate^{8,34}). Pre-gastrulation stage embryos showed 1.06 soL1Rs per 1,000 endogenous point mutations, although we observed only one instance from the 27 individuals. In contrast, the rates were close to 0 soL1Rs per 1,000 endogenous point mutations for the blood and fibroblast lineages, regardless of embryonic and post-embryonic stages (**Fig. 1i**).

Fig. 1i (new). Normalised soL1R rates, estimated by the number of soL1Rs per 1,000 somatic point mutations, in various stages and cell types.

-Issue 1-k) We appreciate the positive comment from the reviewer. We agree with the reviewer that this part is one of the main findings of this study. We first added more whole-genome methylation and transcriptome data from the clones to make a robust conclusion from the multi-omics analyses. The number of normal colorectal clones in which multi-omics investigated increased from 29 to 132 during the revision process. In addition, 7 fibroblast clones also have DNA methylation sequenced to directly compare the L1 promoter methylation without technical variations (previously, we used bulk methylome data from ENCODE). This data enables more comprehensive analyses and provides more robust conclusions. Multiomics from a large number of clones provided a higher statistical power to understand 1) the timing of epigenotype determination (**Fig. 3f**), 2) individual variations in demethylation frequency of an L1 source, and 3) the range of L1 promoter demethylation (**Figs. 3g, 3h**). L1 promoter demethylation was very rare in fibroblast clones, suggesting that the overall processes of the L1 promoter demethylation were specific to the embryonic lineages differentiating into the colorectal epithelium. The methylation landscape of L1 sources is extensively represented in **Fig. 3b** and **Extended Data Figs. 7-9**) in the revised manuscript (not shown here as they are extensive figures).

We have updated #372-394 accordingly. Of note, the parts were moved ahead to highlight these parts in the manuscript.

[Revised manuscript, page 9, Results, Section “Promoter demethylation of rc-L1s”]

To explore the epigenetic foundation of the differential rc-L1 activities in normal cells, we conducted multi-omics analyses by integrating whole-genome DNA methylation (139 clones), RNA expression profiles (116 clones), and early clonal phylogenies of the clones in 9 individuals (**Figs. 1a, 3a**). As reported in bulk tissues^{18,41}, a strong negative correlation between L1 promoter demethylation and transcription from the source elements was observed in the clones (**Extended Data Fig. 6a**).

The frequency of promoter demethylation (and their resultant transcription) varied across cell types and source elements (**Fig. 3b; Extended Data Figs. 7-9**). Although most rc-L1 promoters were methylated in the fibroblast clones, promoter demethylation were remarkably observed in the colorectal clones. For instance, promoter of the prevalent-active 22q12.1-2 source element showed frequent homozygous demethylation (and resultant RNA transcription) in most colorectal clones (**Figs. 3b, 3c**). Another prevalent-active 12p13.32 source element displayed demethylation in a substantial proportion of clones in all individuals (**Figs. 3b, 3d**). Occasionally, rc-L1 promoter demethylation was more prevalent in specific individuals, as observed in 5q14.1-1 and 14q12-3 source elements (**Fig. 3b**). The frequency of the rc-L1 promoter demethylation could be higher in the colorectal epithelium than in other cell types (**Extended Data Fig. 6b**)⁴².

Of note, we observed that population-prevalent rc-L1s were repressed through promoter methylation. Of the 89 previously known population-prevalent rc-L1s, 69 (77.5%) showed promoter methylation with >70% frequency in the colorectal clones. Another 9 (10.1%) without frequent promoter methylation, such as the 12q13.13, showed ORF truncating mutations in the rc-L1 sequences. Intriguingly, the four prevalent-active sources (22q12.1-2, Xp22.2-1, 1p12, and 12p13.32) escaped from both the genetic and epigenetic repressions (**Fig. 3e**), which may indicate the functional roles of the sources⁴³.

Multi-dimensional analysis further provided four insights into the epigenetic regulation of source elements and subsequent soL1R activity in somatic lineages. First, rc-L1 promoter demethylation is a prerequisite condition for soL1Rs. A colorectal clone has on average 17–41 promoter-demethylated rc-L1 alleles in the genome, and the source elements that caused any transduction events were always promoter-demethylated in the corresponding clone (**Fig. 3b**; 47 out of 47; highlighted by red boxes; 36 homozygous and 11 heterozygous demethylations, respectively). On the contrary, promoter demethylation is insufficient for inducing soL1R, as most promoter-demethylated rc-L1 sources did not generate any transductions in clones. This indicates that the rc-L1 promoter epigenotype is stable in the somatic lineages over time, as its dynamic changes would disrupt such an exclusive association.

Second, L1 promoter epigenotype is primarily determined in embryogenesis. Rc-L1 promoter demethylation was predominantly homozygous (**Figs. 3b-3d**), suggesting that it is directly

inherited from the pre-gastrulation epigenetic reprogramming, which globally removes DNA methylations in the genome⁴⁴⁻⁴⁶. An alternative scenario, the stochastic loss-of-methylation in the ageing process, is less likely because it will shape heterozygous demethylation preferentially. Although the remethylation of rc-L1 promoters in the later stage was almost complete in the fibroblast lineages, the process should be insufficient in the colorectal epithelial lineages (**Fig. 3b**). The molecular time in the phylogenies further indicated that the remethylation process was predominantly operative in a certain embryonic period: colorectal clones branched from common ancestral cells that existed in the 17–65 embryonic mutations of molecular time (12th-90th cell generations assuming the abovementioned fixed early mutation rate^{8,34}; the period is near gastrulation to organogenesis) exhibited a higher concordance of the promoter epigenotype for a source element (77% concordance rate; 1,431 out of 1,868 clone-L1 pairs) than did clones diverged in an earlier time period (**Figs. 3b-3d, 3f**). Our data illustrate that L1 promoter epigenotypes are determined in the post-gastrulation embryonic stage and then stably inherited through downstream mitoses in the colorectal lineages, like X-inactivation^{14,47}.

Third, the range of insufficient remethylation of the promoter regions is rc-L1-localised and is independent from the other genomic regions. For example, despite the extreme difference in the promoter of the prevalent-active 22q12.1-2 source between fibroblast and colorectal clones, its 100 kb upstream and downstream regions showed highly similar DNA methylation profiles (**Fig. 3g**). Likewise, DNA methylation levels of the neighbouring and genome-wide regions were mostly concordant in colorectal clones, regardless of the L1 promoter epigenotypes (**Fig. 3h; Extended Data Figs. 10a, 10b**).

Lastly, most L1 transcripts are unproductive regarding soL1Rs in normal cells. Our transcriptome data suggest that a colorectal epithelial lineage is continuously exposed to a few rc-L1 transcripts over a lifetime, given the expression levels (average 0.6 FPKM per clone when aggregated; **Extended Data Figs. 7-9**)⁴⁸. However, on average, 3 soL1Rs accumulate in the lifetime, indicating that the defence mechanism that protects the retrotransposition of L1 transcripts is potent in the normal colorectal epithelium.

Figs. 3e-3h (3e, 3g, 3h new; 3f improved). **e**, Proportion of non-truncated alleles and clones with demethylated promoters in 89 population-prevalent rc-L1s. Red dots indicate four prevalent-active sources. Black and gray dots indicate common sources with any and no

*transduction events in our study, respectively. f, Clones branched from early cells after 17 somatic mutations in molecular time exhibit similar rc-L1 methylation profiles. Differences in methylation levels of rc-L1s in each pair of clones are correlated with the number of early embryonic mutations in their most recent common ancestral cell. Of the 156 rc-L1s present in the germline of 7 individuals, 29 showing a substantial methylation variation across clones are selected for the analysis. The approximate number of mitoses was calculated using the fixed early endogenous point mutation rate from Ref. 8. *P<0.0125 (Two-sample Kolmogorov-Smirnov test). g, Methylation profile of 100 kb upstream and downstream region of rc-L1 at 22q12.1-2 in colorectal and fibroblast clones. The region for rc-L1 is highlighted in yellow boxes. The genomic coordinates of CpG sites are depicted in the top panel. Middle panel shows the fraction of methylated CpG in colorectal (gold) and fibroblast (silver) clones. Bottom panel shows the differences in the fraction of methylated CpG between colorectal and fibroblast clones. mCpG, methylated CpG. h, Methylation profile of 100 kb upstream and downstream region of rc-L1 at 1p12 in colorectal clones with open and closed promoters. The region for rc-L1 is highlighted in yellow boxes. The genomic coordinates of CpG sites are depicted in the top panel. Middle panel shows the fraction of methylated CpG in colorectal clones with open (orange) and closed (blue) promoters. Bottom panel shows the difference in the fraction of methylated CpG between colorectal clones with open and closed promoters.*

Comment 1-6. The methods (molecular biology and computational) employed are cutting-edge, although there is not much innovation. For instance, **1-l**) identification of L1 somatic retrotranspositions is based on a method described in 2014, which can be improved to identify more source elements. **1-m**) The authors are assuming a fixed rate of mutation as a molecular clock, from reference 8, that is later used to perform very precise timing (i.e., cell divisions) of somatic retrotranspositions. I really love it, but I wonder how precise this is...

Answer 1-6

-Issue 1-l) For the most sensitive detection of soL1R events, we combined multiple tools (MELT, TraFic-mem, xTea, and DELLY) and also conducted extensive bioinformatics filtering and final manual curation by two authors independently. Therefore, although we do not have a technical innovation, as the reviewer pointed out, we are sure that our soL1R (and the other mobile elements) calls are both sensitive and specific. It allowed us an accurate measurement of soL1R rates and sophisticated characterization of the inserted sequences in normal cells.

-Issue 1-m) We appreciate the comment. The method is possible when somatic mutations reconstruct an early phylogeny of multiple clones. The reviewer's concern is reasonable as the molecular clock may not be perfectly uniform in every cell division. However, many independent works suggest more-or-less similar early mutation rates (including Coorens et al., Nature 2021; Ju et al., Nature 2017). In addition, the underlying assumption of the overall constant mutation rate in every cell division is biologically plausible as these early mutations are likely caused by stochastic endogenous processes such as 5-methyl cytosine

deamination (SBS 1) and/or spontaneous errors in DNA replication (which should have a certain error rate; SBS 5) (Park et al., Nature 2021). Finally, the method is the only way to measure the timing of early embryonic events in humans, which has been applied in many previous works (Coorens et al., Nature 2021; Chapman et al., Nature 2021; Lee-Six et al., Nature 2018).

To abide by the reviewer's concern and not to mislead readers, we explicitly describe that the timing is with an assumption of a fixed rate of somatic mutations. In addition, in the revised manuscript, we used molecular clocks assessed by two independent studies (Coorens et al., Nature 2021; Park et al., Nature 2021) for translating mutation numbers into cell generations. It provides a broader range of cell generations, although our conclusions are still robust.

[Original manuscript, #202-#205]

Although the event clearly shows that soL1R is possible at the very early stage embryogenesis, the other nine shared soL1Rs by colorectal clones were absent in blood cells. Their latter node positions in the phylogenies and later molecular time (16-56 EEMs, which is equivalent to the 11th-45th cell generations) were consistent with post-gastrulation8 (Fig. 1h; Extended Data Figs. 3, 4).

⇒

[Revised manuscript, page 6, Results, Section "High soL1R activity in embryogenesis"]

*The other nine shared soL1Rs were likely post-gastrulation embryonic events, given the downstream positions and molecular time of their ancestral nodes in the phylogeny (16–56 point mutations of molecular time, which is equivalent to the 11th–78th cell generations assuming the fixed point mutation rate in embryogenesis^{8,34}), and the absence of the soL1Rs in the blood (**Fig. 1h; Extended Data Figs. 3, 4**).*

[Original manuscript, #210-#217]

Of 19 phylogenies by normal colorectal clones, informative embryonic lineages (early shared branches) included a total of 2,827 mutations of molecular time (Extended Data Figs. 3, 4). Therefore, our observation (10 embryonic soL1Rs) suggests one soL1R per 283 somatic point mutations in the early stage of embryonic development (95% CI: one in every 175-744 mutations; equivalent to $1.6 \times 10^{-3} \sim 6.8 \times 10^{-3}$ soL1R per cell per cell division), which was 3.3-fold higher than the rate at the late somatic lineages unique to single clones (one per 940 clock-like mutations in molecular time; $P=0.001$, two-sided Poisson exact test). Our findings indicated that the soL1R rate is higher in embryogenesis than in somatic lineages after birth.

⇒

[Revised manuscript, page 7, Results, Section "High soL1R activity in embryogenesis"]

To compare soL1R rates in various stages and cell types, we normalised the soL1Rs using endogenous point mutation (SBSs 1 and 5). The soL1R rate in the colorectal epithelial lineages after birth was 1.2 soL1Rs per 1,000 endogenous point mutations. The rate was ~4 times higher in the post-gastrulation embryonic stages differentiating to the colorectal lineages (4.52 per 1,000 endogenous point mutations; $P=8.6 \times 10^{-4}$, two-sided Poisson exact

test), which is equivalent to $1.1 \times 10^{-3} - 9.0 \times 10^{-3}$ soL1R per cell per early cell division (assuming the fixed early endogenous point mutation rate^{8,34}). Pre-gastrulation stage embryos showed 1.06 soL1Rs per 1,000 endogenous point mutations, although we observed only one instance from the 27 individuals. In contrast, the rates were close to 0 soL1Rs per 1,000 endogenous point mutations for the blood and fibroblast lineages, regardless of embryonic and post-embryonic stages (**Fig. 1i**).

[Original manuscript, #368-#372]

Indeed, clones branched from common ancestral cells that existed in the 16-192 embryonic mutations of molecular time in the phylogenetic trees (10^{th} - 158^{th} cell generations; a common ancestral cell near gastrulation to organogenesis; the reference timing of human embryogenesis in molecular time is available in Ref. 8) exhibited similar promoter methylation patterns for a specific hot L1 (Figs. 4d, f, g; Extended Data Fig. 7).

⇒

[Revised manuscript, page 10, Results, Section “Promoter demethylation of rc-L1s”]

The molecular time in the phylogenies further indicated that the remethylation process was predominantly operative in a certain embryonic period: colorectal clones branched from common ancestral cells that existed in the 17–65 embryonic mutations of molecular time (12^{th} - 90^{th} cell generations assuming the abovementioned fixed early mutation rate^{8,34}; the period is near gastrulation to organogenesis) exhibited a higher concordance of the promoter epigenotype for a source element (77% concordance rate; 1,431 out of 1,868 clone-L1 pairs) than did clones diverged in an earlier time period (**Figs. 3b-3d, 3f**).

Referee #2 (Remarks to the Author): expertise-cancer evolution, genomics

Comment 2-1 This is an interesting and clearly presented manuscript describing the somatic evolution of somatic line-1 retrotransposition (sol1R) in ageing human cells. By sequencing single cell-derived colonies from patient material, and combining these genomic data with gene expression and DNA methylation data, the authors report the prevalence, rate and clonality of sol1Rs, finding evidence of increased rate of sol1R in embryogenesis and tumorigenesis. There are interesting results that utilise human population scale data that suggest sol1R is generally under negative selection (with implications for understanding how this negative selection is relaxed in carcinogenesis). The final section of the manuscript combines multi-omic data to explore potential molecular regulators of sol1R, finding (perhaps unsurprisingly) that sol1R requires the L1 to be unmethylated and expressed in order to be inserted; this final section is the least developed of the paper. Overall the manuscript offers careful and quite thorough insights into the somatic evolution of L1 elements in normal human tissues.

Answer 2-1

It is a fantastic summary of our work, and we appreciate the reviewer's time and effort in evaluating our manuscript and providing constructive comments. Reviewer 2 mentions that the final section (Fig. 4) is the least developed. In line with our Answers 1-5, the revised manuscript focuses more on the early regulation of retrotransposition-competent L1 (rc-L1). To this end, the section is now moved to the earlier part (to Fig. 2). As described in Answer 1-5, we produced whole-genome methylation and transcriptome sequencing data from an additional 103 colorectal epithelial clones (now 132 colorectal clones are multi-omics profiled). We also sequenced the whole-genome methylation of seven fibroblast clones for a direct comparison. These datasets provide deeper insights into the early molecular regulation of rc-L1 sources.

Comment 2-2 1. I think the sol1R detection in cancers is on bulk tissue (where there are likely many genetically diverged lineages, that each could have a different sol1R) so the comparison of sol1R burden with normal clones should be tempered, or alternatively, a comparison of sol1R burden where cancer cell fraction (CCF) is estimated to be 1 could be compared. This is an important analysis as it supports/refutes the claim that sol1R burden is increased in cancer.

Answer 2-2

We appreciate the reviewer for the reasonable comment. Our initial concern was the potential low sol1R detection sensitivity in cancers due to the contamination of normal cells in tumor tissues. However, we learned that we might find more events in cancers if we detect many subclonal events in tumour tissues. To abide by the reviewer's comment, we estimated the cancer cell fraction (CCF) from all sol1R events in cancers and concluded that ~97% of the cancer sol1Rs are clonal ones. The clonal proportion of sol1R events in

cancers is shown in the revised manuscript, and the soL1R rate in cancers is updated accordingly, as the reviewer suggested. As the vast majority of soL1Rs in cancers are clonal, we revealed that the soL1R rate is still higher in tumourigenesis.

[Revised manuscript, page 5, Results, Section “Widespread soL1R in normal colorectal epithelium”]

*In the 887 normal and 12 MUTYH-associated adenomatous clones, we identified 1,251 and 458 soL1Rs, respectively, by a combined analysis using four different bioinformatics tools (**Extended Data Fig. 1e; Supplementary Tables 1, 2**). We further found 572 soL1Rs from the 19 matched cancers, 97.2% of which (n=556) were clonal events shared by all cancer cells in the tissue. For the other retrotransposon types, we additionally detected 9 somatic Alu insertions in the normal clones (**Supplementary Table 2**).*

[Revised manuscript, page 12, Results, Section “Acceleration of soL1R activity during tumourigenesis”]

*Lastly, we sought to determine the soL1R burden and rate in tumourigenesis. All 19 matched colorectal carcinomas harboured soL1R events with considerable variance ranging between 4 and 105 (**Fig. 1b**). Most soL1R events identified in cancer were clonal (n=556; 97.2%). On average, the number of clonal soL1R events was 29 per cancer, approximately 10 times more frequent than observed in normal colorectal clones.*

Comment 2-3

2-a) What is the interrelationship between copy number alterations (CNAs) and other structural alterations (SAs) and soL1Rs? **2-b)** Could it be that some soL1R events are actually duplications caused by other SAs? (are these the complex events referred to in Figure 2B?) I would think the analysis is robust to this (as the authors examine flanking regions of L1 insertion sites) but it would be good to confirm. **2-c)** Relatedly, do the authors think that the purported higher burden of soL1Rs in colorectal adenomas/cancers is interrelated with tolerance of chromosomal instability that is common in these cancers? (the authors could explicitly assess soL1R burden in MSI vs MSS cases, and/or genome doubled cancers, and repeat the regression in Fig 2d comparing normal vs cancer, perhaps making use of PCAWG cases)

Answer 2-3

These are all constructive comments which make our manuscript more solid. We appreciate the reviewer for these inputs.

-Issue 2-a) Because normal cells do not have frequent copy number alterations and structural variations in the genome, we are underpowered to correlate between soL1Rs and

other structural variations in these non-neoplastic clones. In addition, there is no apparent relationship in colorectal cancers (Extended Data Fig. 12d and PCAWG data).

-Issue 2-b) As the reviewer speculated, because we confirmed flanking sequences for a soL1R, including target site duplications and poly-A tails, we believe the vast majority of soL1R events we found were not directly associated with other structural variations. We have updated our manuscript to clarify it.

[Revised manuscript, page 4, Results, Section “Widespread soL1R in normal colorectal epithelium”]

Because of the two genomic footprints of retrotransposition (the poly-A tail and target site duplication (TSD)), soL1R events could be clearly distinguished from the other genomic rearrangements (Extended Data Figs. 1a, 1b).

Extended Data Figs. 1a, 1b. **a**, A schematic plot describing two genomic footprints of retrotransposition, the poly-A tail and target site duplication. RT body, retrotransposed body; TSD, target site duplication. **b**, The distribution of target site lengths at insertion sites. Positive and negative target site lengths indicate target site duplication and deletion, respectively. soL1R, somatic L1 retrotransposition.

-Issue 2-c) We think that the soL1R rate increases during tumourigenesis. Our data indicate that tumour cells have, on average, ~10 times higher burden of clonal soL1R and ~3 times higher soL1R rate than their matched normal colorectal clones. The acceleration of the soL1R rate in tumour development is also observed in the clonal phylogeny of adenomas (HC22; Fig. 4j in the revised manuscript). However, we did not find a clear positive correlation between soL1R burden and the copy number changes, microsatellite instability, and TP53 mutation status. It implies that the favorable condition for soL1R in cancers should be different from genomic instability for other structural variations. We have updated our manuscript accordingly.

[Revised manuscript, page 12, Results, Section “Acceleration of soL1R activity during tumorigenesis”]

Our findings suggest that certain phenotypic changes in tumour development may provide more permissive conditions for L1 retrotransposition. Qualitatively, soL1Rs in tumour tissues shaped more profound changes, including longer insert lengths (1,031bp vs. 453bp for solo-L1, $P=8.5 \times 10^{-20}$, two-sided t-test; 755bp vs. 615bp for partnered transductions, $P=0.59$, two-sided Wilcoxon rank-sum test; and 530bp vs. 245bp for orphan transductions, $P=0.004$, two-sided t-test; **Extended Data Fig. 12b**) and a higher frequency of the head sequence variations (41.8% vs. 30.0%, $P=1.3 \times 10^{-6}$, two-sided Fisher exact test; **Extended Data Fig. 12c**). However, the soL1R permissive condition may not be equivalent to the classical genome instability in cancers as we did not observe a clear differential soL1R burden according to DNA copy number instability, TP53 inactivating mutations, and microsatellite instability in colorectal cancers¹⁹ (**Extended Data Fig. 12d**).

Extended Data Fig. 12 (mostly new except for EDF 12b). Differences in genomic features of somatic L1 insertion between normal colorectal clones and colorectal cancers. **a**, The soL1R rate is accelerated during tumourigenesis in colorectal lineages. **b**, Distribution of L1 insertion size in normal colorectal clones and colorectal cancers. **c**, Proportion of soL1Rs events with head variations in normal colorectal clones and colorectal cancers. **d**, The number of soL1Rs between colorectal cancers with or without copy number instability, TP53 inactivating mutations, and microsatellite instability. ns, not significant.

[Revised manuscript, page 12, Results, Section “Acceleration of soL1R activity during tumorigenesis”]

Acceleration of soL1R rate during tumour development was observed in the MUTYH-associated adenomatous clones. In the developmental tree of the adenomatous polyps, the soL1R rate increased as lineages were getting closer to carcinoma, with an accumulation of more driver mutations. For example, the soL1R rate in the lineages with three driver mutations (loss-of-function mutations in APC and ARID1A and a gain-of-function mutation in KRAS) was 3-5-fold higher than that in the lineage with no remarkable drivers (**Fig. 4j**).

Fig. 4j (new). Phylogeny of *MUTYH*-associated adenomatous clones with normalised L1 rates in groups of lineages classified by driver mutations. Branch lengths are proportional to the molecular time measured by the number of somatic point mutations. The number of branch-specific sol1Rs and branch-specific driver mutations are described.

Comment 2-4 3. Are there mutational consequences of sol1R insertion at the insertion points (analogous to kataegis)?

Answer 2-4

We measured the genomic distances between each sol1R insertion site to its nearest somatic point mutation and compared the distances with their background. However, we did not observe shorter distances, suggesting that sol1R insertion does not generally induce other point mutations in the vicinity. We have updated our result section with the analysis as shown below. We appreciate the reviewer for this comment.

[Revised manuscript, page 11, Results, Section “Genomic regions of sol1R insertions”]

The multi-omics integration also suggested that sol1R insertions did not induce additional point mutations, gene expression/splicing changes, or DNA methylation alterations in the nearby regions from the insertion point (**Figs. 4d-4f**). We speculate that clones with functionally damaging sol1Rs were negatively selected in normal tissues.

Figs. 4d-4f (new). Distribution of distances to the nearest point mutations (*d*), gene expression level (*e*), and methylation fraction of nearby region (*f*) in colorectal clones with and without L1 insertions. TPM, transcripts per million.

Comment 2-5 (features of insertion points) 4. Related to Figure 2, do features of insertion sites differ through molecular time?

Answer 2-5

In this study, we classified soL1R events into four groups according to molecular time, i.e., (1) pre-gastrulation (n=1), (2) post-gastrulation (n=9), (3) age-related after birth, and (4) found in cancers. The analysis for insertion sites is not possible for (1) pre-gastrulation and (2) post-gastrulation soL1Rs due to the insufficient number of observations. Except for the embryonic events, we compared genomic features between (3) age-related and (4) cancer-related soL1Rs. As shown in Fig. 4b, we do not find any substantial differences, suggesting that characteristics of target sites vulnerable to soL1R do not change during tumourigenesis.

Comment 2-6

2-d) Regarding the multi-omic analysis of soL1Rs, is there evidence that there are genome-wide differences in DNA methylation (and/or expression) in clones with soL1Rs (perhaps further evidence of a “genomic instability tolerant” phenotype)?

2-e) Similarly, is there evidence of changed methylation status (and so changed chromatin status) at insertion sites? (and the authors could speculate on cause/consequence) The analysis as-is is restricted to looking at the molecular features of soL1R sites only; it feels that more interesting analysis could come from these multi-omic data from a more extensive analysis.

Answer 2-6

- **Issue 2-d)** We appreciate the constructive comment, as it made us broaden the scope of this study. We compared the genome-wide (global) methylation profiles between clones with and without soL1Rs, but we did not find any substantial differences. Now we conclude that the acquisition of soL1Rs in the normal colorectal epithelium is a more-or-less stochastic event: every colorectal clone has a lower probability of soL1R event at any given time. The probability is determined during post-gastrulation embryogenesis through local differential demethylation of rc-L1 promoters rather than global differential demethylation. The manuscript has been updated accordingly, as shown below.

[Revised manuscript, page 10, Results, Section “Promoter demethylation of rc-L1s”]

Third, the range of insufficient remethylation of the promoter regions is *rc*-L1-localised and is independent from the other genomic regions. For example, despite the extreme difference in the promoter of the prevalent-active 22q12.1-2 source between fibroblast and colorectal clones, its 100 kb upstream and downstream regions showed highly similar DNA methylation profiles (**Fig. 3g**). Likewise, DNA methylation levels of the neighbouring and genome-wide regions were mostly concordant in colorectal clones, regardless of the L1 promoter epigenotypes (**Fig. 3h**; **Extended Data Figs. 10a, 10b**).

Figs. 3g, 3h (new). **g**, Methylation profile of 100 kb upstream and downstream region of *rc*-L1 at 22q12.1-2 in colorectal and fibroblast clones. The region for *rc*-L1 is highlighted in yellow boxes. The genomic coordinates of CpG sites are depicted in the top panel. Middle panel shows the fraction of methylated CpG in colorectal (gold) and fibroblast (silver) clones. Bottom panel shows the differences in the fraction of methylated CpG between colorectal and fibroblast clones. mCpG, methylated CpG. **h**, Methylation profile of 100 kb upstream and downstream region of *rc*-L1 at 1p12 in colorectal clones with open and closed promoters. The region for *rc*-L1 is highlighted in yellow boxes. The genomic coordinates of CpG sites are depicted in the top panel. Middle panel shows the fraction of methylated CpG in colorectal clones with open (orange) and closed (blue) promoters. Bottom panel shows the difference in the fraction of methylated CpG between colorectal clones with open and closed promoters.

Extended Data Fig. 10 (new). **a**, DNA methylation levels in the regions nearby source

elements and whole-genomes of colorectal clones. **a**, Methylation profiles of 100 kb upstream and downstream regions of 7 reference rc-L1s with variable methylation levels in colorectal clones. The region for rc-L1 is highlighted with yellow boxes. The genomic coordinates and CpG sites are depicted in the top panel. Middle panel shows the fraction of methylated CpG in colorectal clones with open (orange) and closed (blue) promoters. Bottom panel shows the differences in the fraction of methylated CpG between colorectal clones with open and closed promoters. mCpG, methylated CpG. **b**, A scatter plot showing genome-wide methylation levels of normal colorectal clones in each individual.

- **Issue 2-e**) We also conducted the methylation analysis in 405 soL1R insertion sites detected in colorectal clones with whole-genome methylation sequenced. We screened the methylation status of regions 1 kb upstream and downstream from the insertion point in clones with and without the soL1R event. However, we did not find substantial differences in DNA methylation in the vicinity of a soL1R event. We have updated our manuscript accordingly as well.

[Revised manuscript, page 11, Results, Section “Genomic regions of soL1R insertions”]

The multi-omics integration also suggested that soL1R insertions did not induce additional point mutations, gene expression/splicing changes, or DNA methylation alterations in the nearby regions from the insertion point (**Figs. 4d-4f**). We speculate that clones with functionally damaging soL1Rs were negatively selected in normal tissues.

Figs. 4d-4f (new). Distribution of distances to the nearest point mutations (d), gene expression level (e), and methylation fraction of nearby region (f) in colorectal clones with and without L1 insertions. TPM, transcripts per million.

Comment 2-7. (minor) The statement that “Third, we observed a high level of cell-type specificity in the soL1R burden.” as evidence that soL1R isn’t culture related seems weak: presumably culture conditions were different between cell types, and cell types differentially adapt to culture, which could conceivably cause soL1R.

Answer 2-7

We understand the reviewer's point. We modified the sentences as shown below not to mislead readers.

[Original manuscript, #134, Results, Section "SoL1R rates in normal colorectal epithelium"]

In the 880 normal clones, we identified 1,250 soL1Rs using a combined analysis of four different bioinformatics tools (Supplementary Tables 1, 2). Four lines of evidence indicated that most of the detected soL1Rs were true somatic events that accumulated in vivo rather than culture-induced events. First, the VAFs for soL1Rs were distributed approximately 50%, thus were shared by all cells in a clone (Extended Data Fig. 1c). Similarly, in clones established from male donors, soL1Rs in non-pseudo-autosomal regions of chromosome X exhibited approximately 100% VAF. Second, we experimentally confirmed the rate of culture-associated events in 13 pairs of serial single-cell expansions, directly suggesting that >90% of the detected soL1Rs were true in vivo events (Extended Data Fig. 1d). Third, we observed a high level of cell-type specificity in the soL1R burden. Fourth, we found a positive correlation between the soL1R burden and the age of individuals. The third and fourth features are not expected if most soL1Rs were acquired by culture-associated artifacts.

⇒

[Revised manuscript, page 5, Results, Section "Widespread soL1R in normal colorectal epithelium"]

*Three lines of evidence indicated that most of the soL1Rs in the clones were true somatic events rather than culture-induced events. First, the VAFs for soL1Rs in clones were distributed approximately 50%, and thus were shared by all cells in a clone (**Extended Data Fig. 1f**). Similarly, in clones established from male donors, soL1Rs in non-pseudo-autosomal regions of chromosome X exhibited approximately 100% VAF. Second, as we will see, the soL1R burden per clone increased with age, which was not possible if most instances were culture-mediated artifacts. Third, we experimentally confirmed the rate of culture-associated events in colorectal epithelium using 13 pairs of serial single-cell expansions, directly suggesting that >90% of the detected soL1Rs were true in vivo events (**Extended Data Fig. 1g**). In addition, the paucity of soL1Rs in the blood and fibroblast clones implies that cell culture conditions are not a sufficient condition for L1 retrotransposition.*

Comment 2-8. (minor) All the adenomas appear to be from one individual with a germline-caused (MUTYH) mutator phenotype, so these adenomas are unlikely to represent the wider population – this should be acknowledged.

Answer 2-8

We understand the reviewer's point. We explicitly described that these clones were established from a specific adenoma type (MUTYH), and thus may not appropriately represent general adenomas.

[Revised manuscript, page 4, Results, Section “Widespread soL1R in normal colorectal epithelium”]

*To detect soL1R events, we explored 918 whole-genome sequences, including those of clones from healthy tissues (n=887; 27 individuals), adenomatous polyps (n=12; from four polyps of one patient with MUTYH-associated polyposis), and matched colorectal cancer tissues (n=19; 19 individuals; **Fig. 1a**).*

...

*In the 887 normal and 12 MUTYH-associated adenomatous clones, we identified 1,251 and 458 soL1Rs, respectively, by a combined analysis using four different bioinformatics tools (**Extended Data Fig. 1e; Supplementary Tables 1, 2**). We further found 572 soL1Rs from the 19 matched cancers, 97.2% of which (n=556) were clonal events shared by all cancer cells in the tissue....*

[Revised manuscript, page 12, Results, Section “Acceleration of soL1R activity during tumourigenesis”]

Acceleration of soL1R rate during tumour development was observed in the MUTYH-associated adenomatous clones. In the developmental tree of the adenomatous polyps, the soL1R rate increased as lineages were getting closer to carcinoma, with an accumulation of more driver mutations.

Referee #3 (Remarks to the Author): expertise-cancer evolution, genomics

Comment 3-1. Nam and Youk et al present a detailed study on L1 retrotransposition during development from a large number of expanded cellular colonies from healthy adults. Overall the study is detailed and well written, with a really nice narrative throughout the paper. The paper also does a nice job complementing other recent works exploring human lineage reconstruction with a variety of mutational patterns (SNV, WGS+sanger, etc), putting this work into perspective (Evrony, etc). The characterization of new hot L1s and mechanisms is a really nice angle, and the detail throughout the paper is appreciated. The figures are also really well done and visually explain the story well. My major concerns are about the generalizability from the tissues sampled here to the more global view of L1 retrotransposition.

Answer 3-1

It is an excellent summary of our work, and we appreciate the reviewer's comment.

Comment 3-2. One big concern is the disconnect between the promise of L1 retrotransposon characterization in normal tissue vs the limited number of tissues tackled here. The focus is really the colon; one option would be to tone back the language a bit. For instance you say "In this study, we observed a substantial level of somatic mosaicism in normal cells driven by soL1Rs. However, many things are yet to be discovered. For example, some other cell types not investigated in this study may have higher soL1R rates than colorectal epithelium", but many points in the paper speak to the global perspective ("cell types not in this study" is almost all cell types). Certainly the ideal option would be to validate with PCR (or other targeted approach) some of the L1 transpositions you see here in other tissue samples from the same patient, especially to validate the lineage information. But more clearly stated limitations in the discussion would be helpful.

Answer 3-2

These comments are insightful and also constructive. To our knowledge, this work is the first deep analysis of soL1R in non-neoplastic (normal) cells at the absolute single-cell resolution. Because we found a substantial number of soL1R events, we claimed that soL1Rs are indeed present not only in cancer but also in normal cells.

To abide by the reviewer's comment, we checked soL1Rs in other normal cell types, including fibroblasts and haematopoietic stem and progenitor cells. The most extensive work would be achieved by the clonalization and whole-genome sequencing of other types of normal cells, ideally from the same individuals. However, it is not plausible for this work, as sample collection from multiple tissues is not usually possible from the same patients, and clonalization of different cell types needs tremendous work, including setting up cell culture (organoid) conditions. Targeted methods, such as PCR, were possible, but their results

would be provisional as they lack single-cell resolution, and direct comparison of the soL1R rate against clones would be very challenging.

Finally, we decided to analyze soL1Rs from WGS of laser-capture microdissected patches, despite their known limited detection sensitivity (Supplementary Discussion). In 259 LCM patches from 13 different organs, we identified 37 soL1Rs in most cell types. Our finding is now displayed in a new **Fig. 1f**, and we have updated our manuscript accordingly, as shown below. Of note, the soL1R burden detected from these cell types should be interpreted as a lower bound. The concerns about the generalizability of our findings were also described.

[Revised manuscript, page 6, Results, Section “Widespread soL1R in normal colorectal epithelium”]

To glimpse the soL1R burden in other normal cell types, we investigated 259 whole-genome sequences produced from LCM-based patches dissected from 13 organs³⁴ (**Fig. 1f**). Overall, we detected 37 soL1Rs in these organs (**Supplementary Table 3**), implying that the soL1Rs in normal cells are not confined in the colorectal epithelium. However, the soL1R rates in these organs should be interpreted carefully because of the compromised soL1R detection sensitivity from the LCM-based patches (**Supplementary Discussion**). More systematic and larger-scale studies are necessary to understand and compare soL1R burdens in various normal cell types.

Fig. 1f (new). Number of LCM-based patches with various numbers of soL1Rs across different organs. LCM, laser capture microdissection.

Finally, we toned down our claims to abide by the reviewer’s comment. The title of this manuscript is updated.

Original title

Extensive mosaicism by somatic L1 retrotransposition in normal human cells

⇒

Revised title

Extensive mosaicism by somatic L1 retrotransposition in normal colon epithelium

Comment 3-3 (soL1R rates between normal and tumour) At line 179 (thank for line numbers!) you say that tumor cells have rates about 10X higher than normal cells. In figure 1g/h you also detail the huge increase in mutations in the carcinomas, could you detail the L1 rate per inferred molecular time / cell division for the tumors? Sorry if this doesn't make sense, but I'm curious if the increased L1 'rate' is a by-product of the increased cell division count. This goes to the closing point at line 394 as well, as this could affect the conclusion about defense mechanisms (?)

Answer 3-3

We appreciate the reviewer for this insightful comment. In this work, we distinguish the soL1R (mutation) burden (referred to as "the burden") and the soL1R rate (referred to as "the rate"). As the reviewer may feel, the burden means the number of events per cell at a given time, and the rate means the number of new events per cell division (or per endogenous mutation as the proxy of cell divisions). In the revised manuscript, to abide by the comment 2-2, we count clonal soL1R in cancers, and the burden is still ~10-fold higher in colon carcinomas than in normal colon epithelium.

To answer this question, we calculated the soL1R rate in colon carcinomas. Per 1,000 endogenous point mutations, the rate is ~3-fold higher (accelerated) in carcinomas than in normal epithelium. A similar trend, or lineage-specific soL1R rate acceleration, is also observed during adenoma formation, which is now shown in new Fig. 4J. Collectively, we conclude that the higher soL1R burden in carcinomas is not entirely caused by the increased number of cell divisions in cancers.

We have updated our manuscript as shown below:

[Original manuscript, #171, Results, Section "Acceleration of soL1R rates in tumourigenesis"]

Acceleration of soL1R rates in tumourigenesis We compared the soL1R landscape in the normal colorectal clones (1,236 soL1Rs from 406 clones) with those in neoplastic cells, including MUTYH-associated adenomatous polyps (457 soL1Rs from 12 clones) and matched colorectal carcinomas (572 soL1Rs from 19 tissues; Supplementary Table 2). All adenoma clones and carcinoma tissues harbored soL1Rs, more frequently than the normal epithelium (100% vs. 88%; $P=0.037$, two-sided Fisher exact test). The soL1R burden per cell in adenoma and carcinoma was 38 and 30 soL1Rs on average, respectively, with considerable variance, ranging between 2-66 in the 12 adenoma clones and 4-105 in the 19 carcinoma tissues (Fig. 1b). The soL1R burden was approximately 10-fold higher in adenomas and carcinomas compared to normal cells, suggesting that the processes of

neoplastic transformation provide more favorable conditions for L1 retrotransposition processes.

⇒

[Revised manuscript, page 12, Results, Section “Acceleration of soL1R activity during tumourigenesis”]

Lastly, we sought to determine the soL1R burden and rate in tumourigenesis. All 19 matched colorectal carcinomas harboured soL1R events with considerable variance ranging between 4 and 105 (**Fig. 1b**). Most soL1R events identified in cancer were clonal ($n=556$; 97.2%). On average, the number of clonal soL1R events was 29 per cancer, approximately 10 times more frequent than observed in normal colorectal clones. The soL1R rate in colorectal carcinomas was 3.47 per 1,000 endogenous point mutations, which is ~3 times higher than estimated in normal colorectal lineages (**Extended Data Fig. 12a**).

Extended Data Fig. 12a (new). The soL1R rate is accelerated during tumourigenesis in colorectal lineages.

Comment 3-4 (double integration from a soL1R transcript) The double integration is a nice catch! (277)

Answer 3-4

We appreciate the reviewer’s compliment. To highlight the finding, we moved it from Extended Data Fig. 5b to main Fig. 4I. We feel that it is an intriguing observation as well because it indicates that some L1 transcripts induce multiple retrotransposition events, although most are fruitless.

[Revised manuscript, page 12, Results, Section “Mechanisms of soL1Rs inferred from the breakpoints”]

Interestingly, we found two clones, each of which had transductions at different genomic target sites but with exactly the same length of the unique sequences (**Fig. 4i**). Given that poly-A tailing is a random event in the read-through transcription of the L1 downstream

region, our findings suggest that a single L1 transcript can cause multiple retrotransposition events in somatic cells.

Comment 3-5 If you include the four dominant hot L1s in the ultra-rare vs common L1s, I assume this changes the conclusion? It seems a bit strange to exclude these, but maybe this is getting to the next point.

Answer 3-5

To assess the reviewer's comment, we took into consideration additional 85 prevalent L1 sources (population allele frequency > 75%), which were previously known as retrotransposition-competent. As these did not make any somatic retrotransposition events in our cohort, their transduction rates (defined as TPAM, transduction per allele per million endogenous mutations) are 0, similar to other common L1 retrotransposition-competent sources. It suggests that the four sources are true outliers.

To more systematically understand the genomic and epigenomic contexts of these four prevalent-active sources, we performed long-read whole-genome sequencing of two clones using the PacBio platform and whole-genome methylation sequencing from an additional 110 clones (29 in the original manuscript → 139 in the revised manuscript). Our efforts clearly revealed that most of the prevalent L1 sources are repressed via ORF truncating mutations and/or promoter DNA methylation. In contrast, the four prevalent-active sources evade both repressive mechanisms. During the revision, we realized that one of the dominant sources (1p12) has already been pointed out as an outlier in the previous literature (Ebert et al., Science 2021; citation #13 in the revised manuscript). It seems that these four sources maintain their activity despite the negative selection pressure in the human genome. Even though we speculate that their functional roles underlie their persistent activity, how they could escape the negative selection is still unclear.

We have updated our manuscript as shown below:

[Revised manuscript, page 9, Results, Section "Promoter demethylation of rc-L1s"]

Of note, we observed that population-prevalent rc-L1s were repressed through promoter methylation. Of the 89 previously known population-prevalent rc-L1s, 69 (77.5%) showed promoter methylation with >70% frequency in the colorectal clones. Another 9 (10.1%) without frequent promoter methylation, such as the 12q13.13, showed ORF truncating mutations in the rc-L1 sequences. Intriguingly, the four prevalent-active sources (22q12.1-2, Xp22.2-1, 1p12, and 12p13.32) escaped from both the genetic and epigenetic repressions (Fig. 3e), which may indicate the functional roles of the sources⁴³.

Fig. 3e (new). Proportion of non-truncated alleles and clones with demethylated promoters in 89 population-prevalent rc-L1s. Red dots indicate four prevalent-active sources. Black and gray dots indicate common sources with any and no transduction events in our study, respectively.

Comment 3-6 (size of demethylated loci) You have the mapping locations for the 12p13.32 hot L1, which has nearly complete demethylation of the promoter across the clones. Could you fill in more details when you speculate “have crucial functional roles or are located in genomic loci that are difficult to repress.” (331)? This seems important given the detail you put into methylation for some of the other L1s. I worry some people will see the methylation story as very dependent on the individual L1, and it would be nice to have a more detailed story about why these escape the typical pattern (again to the points made around ~381).

Answer 3-6

As the reviewer suggested, we checked the range of promoter demethylation for particular sources whose promoter epigenotypes are variable across clones. For example, the 12p13.32 source is a good candidate for this analysis, as the source shows all possible epigenotypes (i.e., homozygous demethylation, heterozygous demethylation, and homozygous methylation). As mentioned above, we added 110 whole-genome methylation sequencing during the revision process, and now we have a higher statistical power.

In our analysis, the range of demethylation is primarily limited to the L1 promoters. We did not observe an extended demethylation from the promoter. Therefore, we now conclude that the epigenotypes of retrotransposition-competent L1 sources are determined independently from the epigenotypes of their vicinity regions, and our previous speculation (“We speculate that they either have crucial functional roles or are located in genomic loci that are difficult to repress”) is not plausible anymore. We have added our analyses on Fig. 3h and Extended Data Fig. 10 and have updated our manuscript as below. We sincerely appreciate the reviewer’s suggestion that made this meaningful work possible.

Third, the range of insufficient remethylation of the promoter regions is rc-L1-localised and is independent from the other genomic regions. For example, despite the extreme difference in the promoter of the prevalent-active 22q12.1-2 source between fibroblast and colorectal clones, its 100 kb upstream and downstream regions showed highly similar DNA methylation profiles (**Fig. 3g**). Likewise, DNA methylation levels of the neighbouring and genome-wide regions were mostly concordant in colorectal clones, regardless of the L1 promoter epigenotypes (**Fig. 3h**; **Extended Data Figs. 10a, 10b**).

Figs. 3g, 3h (new). **g**, Methylation profile of 100 kb upstream and downstream region of rc-L1 at 22q12.1-2 in colorectal and fibroblast clones. The region for rc-L1 is highlighted in yellow boxes. The genomic coordinates of CpG sites are depicted in the top panel. Middle panel shows the fraction of methylated CpG in colorectal (gold) and fibroblast (silver) clones. Bottom panel shows the differences in the fraction of methylated CpG between colorectal and fibroblast clones. mCpG, methylated CpG. **h**, Methylation profile of 100 kb upstream and downstream region of rc-L1 at 1p12 in colorectal clones with open and closed promoters. The region for rc-L1 is highlighted in yellow boxes. The genomic coordinates of CpG sites are depicted in the top panel. Middle panel shows the fraction of methylated CpG in colorectal clones with open (orange) and closed (blue) promoters. Bottom panel shows the difference in the fraction of methylated CpG between colorectal clones with open and closed promoters.

Extended Data Fig. 10 (new). DNA methylation levels in the regions nearby source elements and whole-genomes of colorectal clones. **a**, Methylation profiles of 100 kb upstream and downstream regions of 7 reference rc-L1s with variable methylation levels in colorectal clones. The region for rc-L1 is highlighted with yellow boxes. The genomic coordinates and CpG sites are depicted in the top panel. Middle panel shows the fraction of methylated CpG in colorectal clones with open (orange) and closed (blue) promoters. Bottom panel shows the differences in the fraction of methylated CpG between colorectal clones with open and closed promoters. **b**, A scatter plot showing genome-wide methylation levels of normal colorectal clones in each individual.

Reviewer Reports on the First Revision:

Referees' comments:

Referee #1 (Remarks to the Author):

All my main concerns have been addressed in this version of the manuscript. I believe the discoveries here have a broad interest for the field of somatic evolution.

Referee #2 (Remarks to the Author):

The authors have responded very well to my previous comments and the new data and analyses are welcome, and generally very convincing. Details comments below:

2-2 OK – these new clonality data on sol1R VAFs in cancer are welcome and convincing.

2-3 Fig S12d(i) shows sol1R burden by CNA burden. I would perhaps be more informative to show this as a correlation (rather than boxplots) as the CNA burden can vary greatly between cancers. The (new) conclusions from these analysis (that there is no relation between structural alterations and sol1R) seem very strong given that they are based on just this boxplot: its hard to prove a negative but the data do look a bit underpowered here (the highest sol1R burdens are all in the high CNA group and these highest burdens are 50% of the data in that group). Also the association with driver mutation burden (fig 4j) is circumstantial evidence that a relationship with CNA burden isn't unreasonable to expect. Could power be boosted here by inclusion of say PCAWG and Hartwig datasets? I also think the authors should consider a multi-variate regression including TP53, CNAs and MSI simultaneously.

2-4 OK – convincing new analysis

2-5 OK

2-6 Thanks the new analysis and data is welcome and convincing.

Minor comments all appropriately responded to.

Referee #3 (Remarks to the Author):

Thank you to the authors for the revised manuscript; I believe it strengthens many of the weaker spots in the original paper. The expansion into a number of new tissues using laser capture microdissection addresses my most significant concern; certainly, there are caveats to this approach (which the authors note), but this broadens the applicability of the manuscript and makes the authors' larger claims more warranted. The more detailed exploration of the methylation status and

its effect on the reactivation of L1s is also an excellent enhancement to the paper. Overall I am satisfied with the changes made. I understand the concerns of reviewer 1 (the work's novelty, noted in Comment 1-3), though this paper provides a nice step forward in our understanding of retrotransposon rates in normal tissues. It would be good if the authors could satisfy the reviewer's concerns, given that this work would be published in Nature.

Author Rebuttals to First Revision:

Referees' comments:

Referee #1 (Remarks to the Author):

Comment 1-1. All my main concerns have been addressed in this version of the manuscript. I believe the discoveries here have a broad interest for the field of somatic evolution.

Answer 1-1.

We are glad that Referee #1 is satisfied with our revised manuscript. We are grateful to the reviewer for the constructive comments that made our manuscript more meaningful.

Referee #2 (Remarks to the Author):

Comment 2-1. The authors have responded very well to my previous comments and the new data and analyses are welcome, and generally very convincing. Details comments below:

Answer 2-1.

We are glad that Referee #2 is also positive with our previous responses.

Comment 2-2. OK – these new clonality data on sol1R VAFs in cancer are welcome and convincing.

Answer 2-2.

We appreciate for the comment.

Comment 2-3. Fig S12d(i) shows sol1R burden by CNA burden. I would perhaps be more informative to show this as a correlation (rather than boxplots) as the CNA burden can vary greatly between cancers. The (new) conclusions from these analysis (that there is no relation between structural alterations and sol1R) seem very strong given that they are based on just this boxplot: its hard to prove a negative but the data do look a bit underpowered here (the highest sol1R burdens are all in the high CNA group and these highest burdens are 50% of the data in that group). Also the association with driver mutation burden (fig 4j) is circumstantial evidence that a relationship with CNA burden isn't unreasonable to expect. Could power be boosted here by inclusion of say PCAWG and Hartwig datasets? I also think the authors should consider a multi-variate regression including TP53, CNAs and MSI simultaneously.

Answer 2-3.

We appreciate the reviewer for the comment. As the reviewer requested, we have updated the previous Extended Fig 12d (which is Extended Figs 12d-12g). Now, we can understand the correlation between the sol1R burden and the structural variation burden in cancer genomes. As the reviewer feels, there seems to be a weak positive relationship in our

dataset (analysis using 19 colorectal cancers; EDF 12d; $P=0.067$). However, if we increase our sample size with PCAWG (+2,677 cancers), the correlation becomes less significant in colorectal cancers (EDF 12e; $P=0.43$). The correlation is not fully linear because there are additional factors that are necessary for the induction of soL1Rs, such as the L1 promoter demethylation. In addition, we conducted multivariate regression analyses, which can be found in the rightmost column in EDF 12d-12g. At the pan-cancer level, tumour histology types (such as 'esophageal cancer'-which may be related to the levels of L1 promoter demethylation) are more critical for the soL1R burden, although the CNA burden is also significant. To show the relationship in many tumour histology types more clearly, we added EDF 13 in the revised manuscript. As the reviewer suggests, CNA was a significant factor for soL1R in a few cancer types, such as esophageal adenocarcinomas, head-and-neck cancers, renal cell carcinomas, and pancreatic adenocarcinomas, to name a few. We hope that our additional analyses are helpful for readers who wish to understand the relationship in human cancer.

We have updated our manuscript, as shown below.

[Original manuscript (page12)]

However, the soL1R permissive condition may not be equivalent to the classical genome instability in cancers as we did not observe a clear differential soL1R burden according to DNA copy number instability, *TP53* inactivating mutations, and microsatellite instability in colorectal cancers¹⁹ (**Extended Data Fig. 12d**).

→

[Revised manuscript (page 11)]

Our findings suggest a permissive condition for L1 retrotransposition in tumour development, not necessarily equivalent to the classical genome instability in cancers. For example, *TP53* inactivating mutations, microsatellite, and chromosomal instability did not show a robust correlation with soL1R burdens in colorectal cancers (**Extended Data Figs. 12d, 12e**). Although chromosomal instability was significant in pan-cancers encompassing >2,600 cancer cases¹⁷ (**Extended Data Figs. 12f, 12g**), the association was weak and not consistent in each tumour histologic type (**Extended Data Fig. 13**).

(updated) Extended Data Fig. 12 | Differences in genomic features of somatic L1 insertion between normal colorectal clones and colorectal cancers. d-g. The number of solLRs between colorectal cancers with or without TP53 inactivating mutations (left), microsatellite instability (middle) and genomic instability (chromosomal instability; right). Sample numbers are shown in parentheses. P values from two-sided t-test (left, middle) and linear regression (right) were shown. P values from multivariate regression were represented in the right space. ns, not significant. **d.** In 19 matched colorectal cancer tissues. **e.** In 19 matched colorectal cancer tissues and 52 PCAWG colorectal cancer tissues. **f.** In 19 matched colorectal cancer tissues and 4 cancer types (colorectal adenocarcinomas, esophageal adenocarcinomas, lung squamous cell carcinomas, and head-and-neck squamous cell carcinomas) showing a higher solLR burden among 40 histologic types in PCAWG. **g.** In 19 matched colorectal cancer tissues with all whitelist PCAWG samples.

(new) Extended Data Fig. 13 | The number of solLRs in cancer types according to the TP53 inactivating mutations and chromosomal instability. The PCAWG whitelist samples (n=2,677) and 19 matched colorectal cancers in this study were combined and analyzed. Cancer types with less than 10 cases were not considered. **a**, Somatic TP53 inactivating mutations and the number of solLR events. Number of cases in each histology type were shown in parentheses. P values from two-sided t-test were shown. NA, not available. **b**, Linear regression between the chromosomal instability (genomic rearrangements) and the number of solLR events in each cancer type. R-squared and P value from linear regression were represented in each panel.

Comment 2-4 – 2-6.

2-4 OK – convincing new analysis

2-5 OK

2-6 Thanks the new analysis and data is welcome and convincing.

Minor comments all appropriately responded to.

We are glad that our previous revisions were meaningful for Referee #2.

Referee #3 (Remarks to the Author):

Thank you to the authors for the revised manuscript; I believe it strengthens many of the weaker spots in the original paper. The expansion into a number of new tissues using laser capture microdissection addresses my most significant concern; certainly, there are caveats to this approach (which the authors note), but this broadens the applicability of the manuscript and makes the authors' larger claims more warranted. The more detailed exploration of the methylation status and its effect on the reactivation of L1s is also an

excellent enhancement to the paper. Overall I am satisfied with the changes made. I understand the concerns of reviewer 1 (the work's novelty, noted in Comment 1-3), though this paper provides a nice step forward in our understanding of retrotransposon rates in normal tissues. It would be good if the authors could satisfy the reviewer's concerns, given that this work would be published in Nature.

Answer 3-1.

We are glad that Referee #3 thinks our manuscript is suitable to be published in Nature. We are grateful to the reviewer for his/her constructive comments.

Reviewer Reports on the Second Revision:

Referees' comments:

Referee #2 (Remarks to the Author):

Thanks for the thorough response to my queries about the relationship between L1R and SCNAs - the new data are a welcome addition to what is a very nice paper.

I've no further comments.

Author Rebuttals to Second Revision:

Referees' comments:

Referee #2 (Remarks to the Author):

Comment 1-1. Thanks for the thorough response to my queries about the relationship between L1R and SCNAs - the new data are a welcome addition to what is a very nice paper. I've no further comments.

Answer 1-1.

We are glad that Referee #2 is satisfied with our revised manuscript.